# Hepatic ASPG-mediated lysophosphatidylinositol catabolism impairs insulin signal transduction

Feiyan Li [1], Hua-Sheng Huang[2], Qingwen Zhao [3], Wei Zhang[1], Ting Shi [1], Wenjing Lv[1], Qi Zhu[1], Haojie Liu [1], Yingjiang Xu[4], Haiyan Huang[1], Qi-Qun Tang [1], Yue Gao[3], Tao Peng[2] & Dongning Pan [1,5 ✉]

## Abstract

**Bioactive glycerolysophospholipids (GLPs) are implicated in the pathogenesis of metabolic dysfunction-associated steatotic liver disease (MASLD) and obesity; however, the mechanisms underlying glycerolysophospholipid-mediated changes in insulin signaling remain poorly understood. Here, we identify the amino acid-metabolism enzyme asparaginase (ASPG) as a critical regulator of systemic fatty acid handling and insulin signal transduction. Hepatic ASPG expression levels negatively correlate with insulin sensitivity in humans. Loss of *Aspg* in MASLD mice modifies the liver hepatokine secretome, enhancing systemic insulin sensitivity. Notably, ASPG bears lysophospholipase activity towards the bioactive lipid lysophosphatidylinositol (LPI) in vitro and in vivo. Mechanistically, *Aspg* deficiency results in accumulation of intracellular LPIs and consequently in suppression of tyrosine phosphatase PTP1B activity. This in turn decreases FOXO1-dependent expression of the hepatokine *Sepp1*, leading to reduced SEPP1 secretion and extrahepatic insulin-sensitization. In summary, this study uncovers a novel biological mechanism whereby ASPG-controlled bioactive lipid levels modulate insulin resistance and insulin secretion, suggesting complementary therapeutic strategies for the improvement of systemic glucose homeostasis.**

**Keywords** ASPG; Lysophospholipase; Lysophosphatidylinositol; PTP1B; Insulin Signaling
**Subject Categories** Metabolism; Molecular Biology of Disease

## Introduction

The liver represents an essential organ in systemic lipid and glucose metabolism, substantially achieving the developmental impact of insulin resistance and type 2 diabetes (T2D). Ectopic lipid accumulation in hepatocytes, leading to metabolic dysfunction-associated steatotic liver disease (MASLD), is significantly correlated with an approximately doubled risk of incident T2D (Targher et al, 2021). Numerous bioactive lipid metabolites, such as increased levels of diacylglycerols (DAGs) and ceramides, are likely to induce hepatic insulin resistance (Raichur et al, 2014; Turpin et al, 2014; Ter Horst et al, 2017; Lyu et al, 2020). Further research is advocated to better establish whether other species of lipid mediator might impact on hepatic as well as systemic insulin signal transduction.

Glycerolysophospholipids (GLPs) consist of a glycerol backbone, a polar phosphate head group at the sn-3 position, a hydroxyl group at the sn-2 (or sn-1) position, and a fatty acyl chain on the sn-1 (or sn-2) position (Drzazga et al, 2014). They are derived from the diacyl-phospholipids that are hydrolyzed one fatty acid by phospholipase A (PLA), and are degraded by lysophospholipase. The specific structure of GLPs authorizes them to fit into the pockets of some G protein-coupled receptors (GPCRs), so that they work as ligands regulating intracellular signaling pathways (Kaffe et al, 2024). Patients with MASLD have higher level of lysophosphatidylcholine (LPC) in livers which is positively correlates with disease severity (Puri et al, 2007; García-Cañaveras et al, 2011). LPC inhibits mitochondrial oxidation rate and induces necroinflammatory injury of hepatocytes (Han et al, 2008; Hollie et al, 2014). Extracellular lysophosphatidic acid (LPA) is produced by Autotaxin (ATX) and manifests signaling activities through binding to lysophosphatidic acid receptors (LPARs). Increased LPA in hepatocyte-specific overexpression of ATX mice promotes hepatic steatosis and fibrosis (Kaffe et al, 2017). Lysophosphatidylinositol (LPI) is identified as the endogenous ligand of GPR55 (Makide et al, 2014). Liver GPR55 expression was increased in both MASLD patients and mice. LPI supplementation to mice and cultured hepatocytes increased fatty acid synthesis, decreased β-oxidation, and activated hepatic stellate cells in a GPR55-dependent manner (Fondevila et al, 2021). In addition to activate GPR55, LPI also has non-receptor mediated effects in endothelial cells and cancer cells (Monet et al, 2009; Bondarenko et al, 2011).

Given the bioactive GLPs can modulate the pathogenesis of chronic liver diseases, we paid a particular attention to a liver-enriched lysophospholipase ASPG (Asparaginase), and explored

[1]Key Laboratory of Metabolism and Molecular Medicine of the Ministry of Education; Department of Biochemistry and Molecular Biology, School of Basic Medical Sciences; Fudan University, 200032 Shanghai, China. [2]Department of Hepatobiliary Surgery, the First Affiliated Hospital of Guangxi Medical University, 530021 Nanning, China. [3]Zhejiang Key Laboratory of Traditional Chinese Medicine for the Prevention and Treatment of Senile Chronic Diseases, Affiliated Hangzhou First People's Hospital, School of Medicine, Westlake University, 310006 Hangzhou, China. [4]Department of Interventional Vascular Surgery, Binzhou Medical University Hospital, Binzhou, Shandong, China. [5]Department of Endocrinology and Metabolism, Qingpu Branch of Zhongshan Hospital, Fudan University, 200032 Shanghai, China. ✉E-mail: dongning.pan@fudan.edu.cn

the physiopathological role of ASPG and its substrate lysopho-spholipids in the control of insulin signal transduction. The ASPG protein comprises an N-terminal bacterial-type cytoplasmic asparaginase-like domain and a C-terminal ankyrin repeat domain (Karamitros and Konrad, 2014). Mammalian ASPG has been shown to lack of asparaginase activities (Pavlova et al, 2018), but harbors lysophospholipase activities in vitro (Sugimoto et al, 1998; Menniti et al, 2010). In this study, we found hepatic ASPG protein levels were positively associated with insulin resistance indexes in MAFLD patients. Adeno-associated virus (AAV)-mediated over-expression of ASPG established a role for highly expressed ASPG in aggravating systemic insulin signal transduction dysfunction. *Aspg* deficiency in hepatocytes improved insulin sensitivity in multiple metabolic tissues and enhanced insulin secretion. The protective effect was owing to drastically increased LPIs in hepatocytes upon *Aspg* deletion, which inhibited protein tyrosine phosphatase 1B (PTP1B) activity, and subsequently suppressed hepatokine *Sepp1* (Selenoprotein P) expression. Our findings provide clear evidence that hepatic ASPG serves as a boost motor for defective insulin signal transduction in the setting of MASLD, highlighting decreasing ASPG expression level as a potential procedure to enhance systemic glucose homeostasis.

# Results

## The liver ASPG level is positively correlated with insulin resistance indexes

Insulin resistance is a critical pathogenic factor for MASLD. Bioactive lysophospholipids are potent modulators of chronic liver diseases (Kaffe et al, 2024). To gain insight into the role of intrahepatic lysophospholipids and their metabolic enzymes in the regulation of insulin sensitivity, we obtained human liver samples from hepatic surgery patients and measured the correlations between homeostasis model assessment of insulin resistance (HOMA-IR) and a panel of (lyso)phospholipases (Fig. EV1). The mRNA levels of *Aspg*, *Pnpla7* (patatin-like phospholipase domain-containing 7) and *Pla2g6* (phospholipase A2 group VI) were strongly correlated with an increased HOMA-IR (Figs. 1A and EV1). Among them, hepatic *Pnpla7* functions as a lysophosphatidylcho-line hydrolase (Heier et al, 2017) and regulates very-low-density lipoprotein (VLDL) secretion (Wang et al, 2020). *Pla2g6* deficiency attenuates hepatic steatosis in obese mice and is identified as one of the MAFLD modifier genes (Deng et al, 2016; Otto et al, 2019). However, *Aspg* is a poorly annotated lysophospholipase that attracted our curiosity. Further analysis revealed that the expression of *Aspg* in livers was positively correlated with triglyceride glucose (TyG) index, another surrogate marker of insulin resistance, but not related to body mass index (BMI) (Fig. 1B,C). Even in diagnosed MASLD livers, the level of *Aspg* was much higher in the HOMA-IR more than 2 subgroup than in the HOMA-IR less than 2 subgroup, and showed a positive correlation with TyG index as well (Fig. 1D–F). These results indicate that the hepatic ASPG expression may be deeply involved in the pathologies of human MASLD-associated insulin resistance.

Next, we checked the hepatic *Aspg* level in the mouse model. As shown in Fig. 1G, ASPG is broadly expressed in various mouse tissues, and the liver showed the most abundant level. The

expression of hepatic ASPG was further elevated in both high-fat-diet-modeled and genetically insulin-resistant mice (Fig. 1H,I). Since ASPG was reported to be a lysophospholipase, we wondered whether it would localize to the endoplasmic reticulum (ER) or lipid droplets, as many of the lipases did. In fact, ASPG dispersed in both the cytoplasm and nucleus as revealed by immunofluorescence staining. It largely overlapped with DIOC6(3) dye that was a fluorescent ER probe, but did not associate with intracellular lipid droplets (Fig. 1J). The western blot further validated ASPG presented in the ER and nucleus fractions of mouse liver (Fig. 1K). Therefore, the data from both human and mouse models imply a prospective interaction relating aberrantly increased hepatic ASPG to insulin resistance.

## FOXO1 upregulates ASPG expression

To explore the molecular pathway contributing to upregulation of ASPG levels in the livers of insulin-resistant mice, we fed mouse primary hepatocytes with palmitate or insulin to manipulate intracellular insulin signaling. We observed a marked increase in both *Aspg* mRNA and protein levels by palmitate treatment, whereas insulin supplement downregulated its level (Figs. 2A and EV2A). In light of the repression of ASPG expression by insulin in hepatocytes, we detected hepatic ASPG protein levels in the fasting and refeeding mice, which showed fluctuation in insulin secretion physiologically (Fig. EV2B). As expected, fasting for 24 h with low serum insulin levels elevated ASPG protein level, which was eventually restored by a 2 h-refeeding (Fig. 2B).

FOXO1 is a transcriptional factor that the insulin-AKT axis provides the most important inhibitory input to its activity by phosphorylation and subsequent degradation (Matsuzaki et al, 2003; Calissi et al, 2021). Upon insulin resistance, AKT-mediated FOXO1 inactivation is attenuated, and FOXO1 activity is augmented (Teaney and Cyr, 2023). We performed FOXO1-binding motif scan within the mouse *Aspg* 2 kb promoter region, and found there was a canonical binding site (GTAAACA) (Shin et al, 2012) located between the −887 to −893 locus and another two potential sites around −1.6 kb and −1.9 kb. Chromatin immunoprecipitation (ChIP) assay revealed endogenous FOXO1 associated with the *Aspg* promoter around −900 region, whereas insulin treatment detached FOXO1 from there (Fig. 2C). Luciferase assay further verified that FOXO1 activated both human and mouse *Aspg* promoters in a dose-dependent manner. The mutation of the FOXO1-binding site or insulin treatment abolished FOXO1 transcriptional activation toward the *Aspg* promoter (Figs. 2D and EV2C). Moreover, insulin treatment completely diminished ASPG induction by overexpressing FOXO1, and knocking down FOXO1 downregulated ASPG in the primary hepatocytes (Fig. 2E,F). These results demonstrate that insulin represses ASPG expression by phosphorylating FOXO1.

## Loss of *Aspg* in the liver improves insulin signal transduction in HFD-fed mice

To unveil the metabolic role of ASPG in vivo, we generated a mouse strain with hepatocyte-specific deletion of *Aspg* (*Aspg*flox/flox with *Alb*-Cre; namely *Aspg* LKO). *Albumin*-Cre recombinase diminished the ASPG protein level by 90% in livers without disturbance to

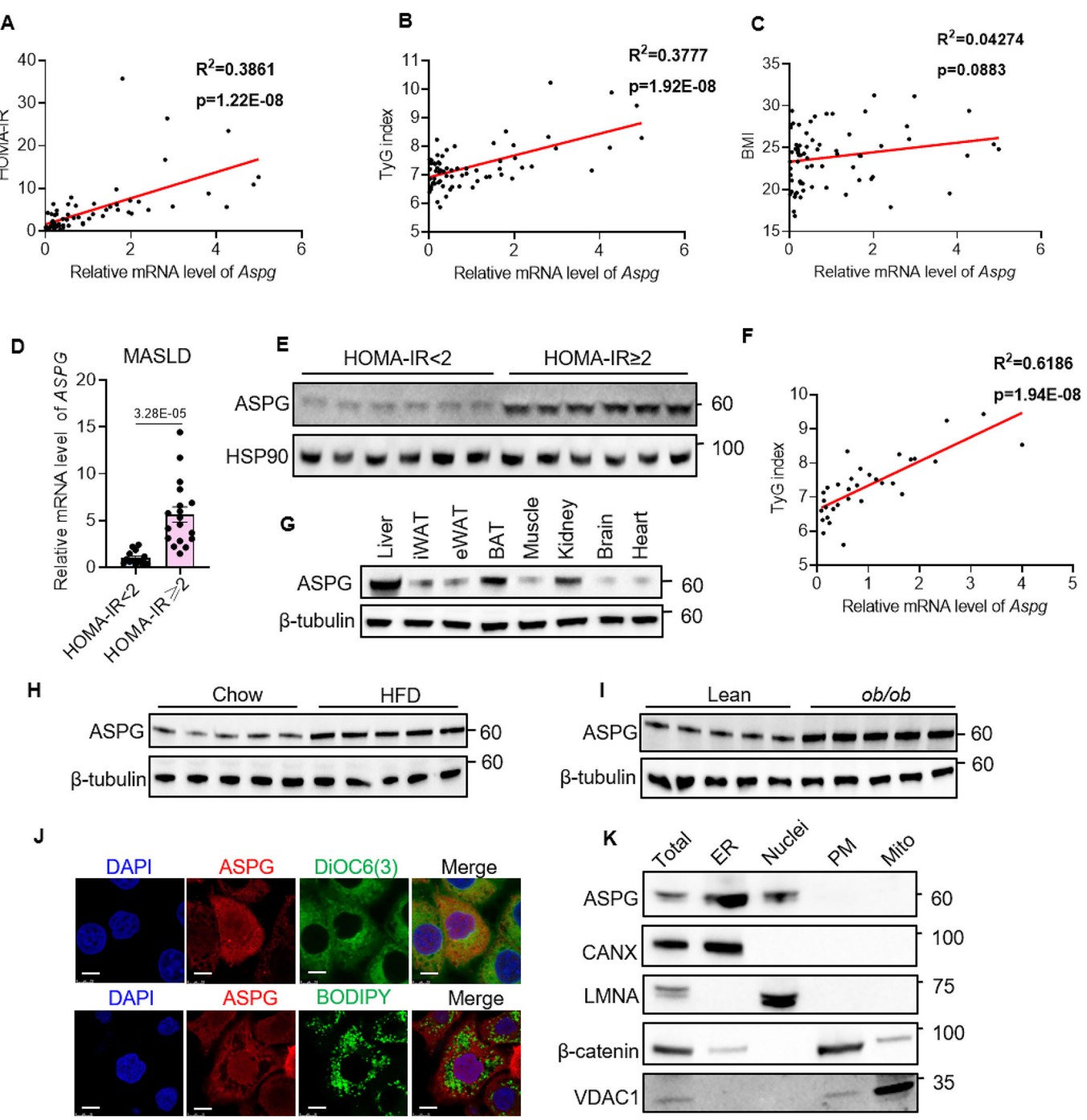

**Figure 1. ASPG upregulation in the liver is correlated with insulin resistance.**

(A–C) The correlation between hepatic *Aspg* mRNA levels and HOMA-IR (A), TyG index (B) and body mass index (BMI) (C) in human liver specimens ($n = 69$). (D, E) Hepatic *Aspg* mRNA (D) and protein levels (E) in the patients with MASLD (HOMA-IR < 2, $n = 14$; HOMA-IR > 2, $n = 18$ for D; $n = 6$ per group for E). (F) The correlation of hepatic *Aspg* mRNA level and TyG index in the patients with MASLD ($n = 32$). (G) Western blot detected ASPG protein expression in mouse tissues. (H, I) Hepatic ASPG protein levels in male chow- and high-fat diet (HFD)-fed mice (H), or in male lean and *ob/ob* mice (I) ($n = 5$ each group for (H, I), male). (J) Representative images of ASPG immunofluorescence staining in HepG2 cells. The ASPG-Flag plasmid was transfected to HepG2 cells, and Flag antibody was used to detect ASPG. The lower panel of cells were treated with 200 μM palmitate for 24 h to induce lipid accumulation. DiOC6(3): dye for endoplasmic reticulum; BODIPY: lipid droplet dye; scale bar 10 μm. (K) Protein levels of ASPG in subcellular fractions of mouse liver tissues. ER endoplasmic reticulum, PM plasma membrane, Mito mitochondria, CANX calnexin, LMNA lamin A/C, VDAC1 voltage-dependent anion channel 1. Data are represented as mean ± SEM. The Pearson and Spearman correlation analysis was applied for (A–C, F). Two-tailed *t* test was applied for (D). Source data are available online for this figure.

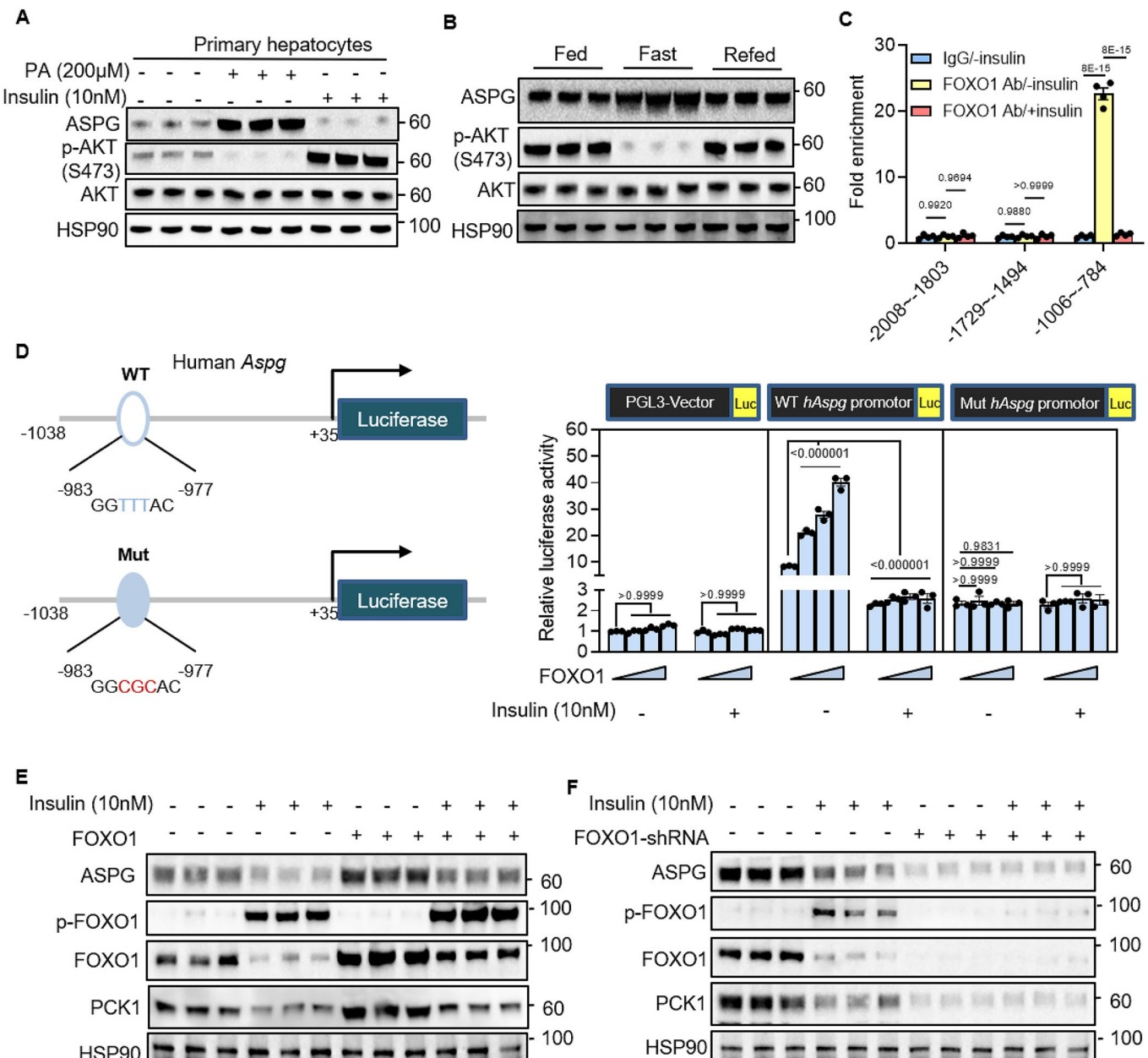

**Figure 2. Insulin represses hepatic ASPG expression by phosphorylating FOXO1.**

(A) Primary hepatocytes isolated from the C57BL/6 mice were treated with palmitate (PA) or insulin for 24 h. The indicated protein levels were analyzed by western blot ($n = 3$ each group). (B) The indicated proteins were detected in livers from feeding, 24 h-fasting and 2 h-refeeding C57BL/6 mice ($n = 3$ each group, male). (C) Chromatin immunoprecipitation (ChIP) analysis on the *Aspg* promoter in primary hepatocytes of C57BL/6 mice ($n = 4$ each group). (D) Dual-luciferase reporter assay using 1 kb human *Aspg* promoter was performed in HEK293T cells ($n = 3$ each group). The potential FOXO1-binding sequence was mutated as indicated. (E, F) Lentivirus-mediated *Foxo1* overexpression (E) or knockdown (F) in mouse primary hepatocytes followed by western blot analysis ($n = 3$ each group). PCK1 was used as a positive control of the FOXO1 target gene. The cells were treated with or without 10 nM insulin for 24 h before the assay for (C–F). Data are represented as mean ± SEM. Statistical analysis was performed by two-way ANOVA followed by Tukey's test for (C, D). Source data are available online for this figure.

ASPG expression in other tissues (Fig. EV3A–C). The adult *Aspg* LKO mice displayed normal liver histology, similar levels of key enzymes engaged in glucose and lipid metabolism, and alike serum glutamic-pyruvic transaminase (ALT) and glutamic-oxaloacetic transaminase (AST) levels as compared with the littermate controls (*Aspg*^flox/flox^; namely WT) under a chow diet (Fig. EV3D–F). Interestingly, the *Aspg* LKO mice had a relatively lower

postprandial blood glucose level, but a higher postprandial serum insulin level than the WT mice (Fig. EV3G,H). Correspondingly, the phosphorylation of the key components in the insulin signaling pathway was significantly enhanced in the *Aspg* LKO mice livers (note, here the western blot was performed in the liver tissues from 2h-refed mice after overnight fasting to induce post-meal insulin surges) (Fig. EV3I). However, the strengthened insulin signaling

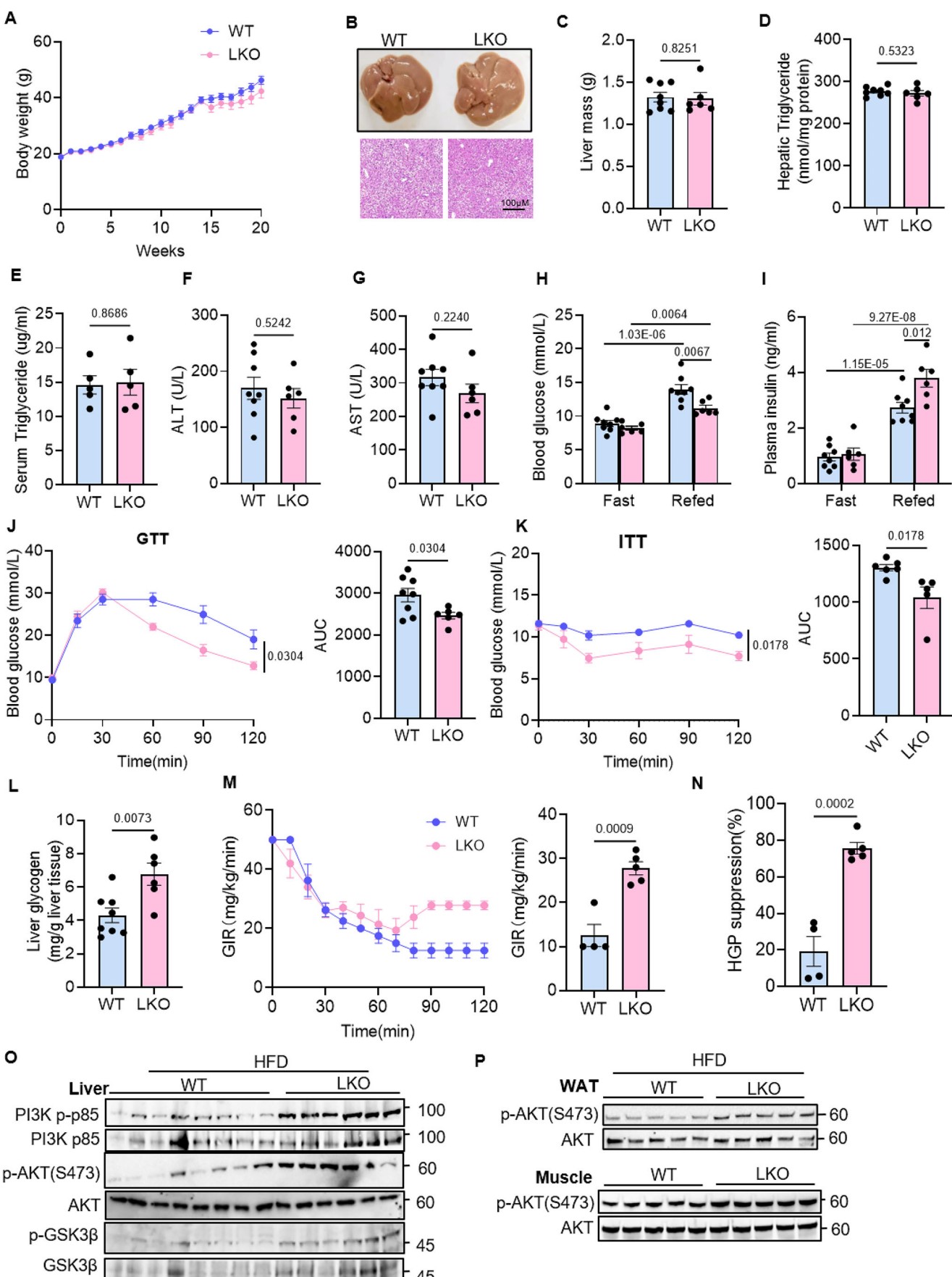

**Figure 3. Hepatic APSG deficiency alleviates obesity-associated insulin signaling defects.**

(A–G) Comparison of the phenotypic features of the WT and *Aspg* LKO male mice after a HFD feeding for 20 weeks, including (A) body weight, (B) liver appearance and H&E staining (scale bars 100 μm), (C) liver weight, (D) hepatic triglyceride, (E) serum triglycerides, (F) serum ALT, and (G) serum AST levels (WT $n = 8$, LKO $n = 6$, male in (A–D, F, G); $n = 5$ each genotype, male in E). (H, I) Levels of blood glucose (H) and plasma insulin (I) in the HFD-fed WT and *Aspg* LKO mice after 24 h-fasting and 2 h-refeeding (WT $n = 8$, LKO $n = 6$, male). (J, K) Glucose tolerance tests (GTT, WT $n = 8$, LKO $n = 6$, male) and insulin tolerance tests (ITT, WT $n = 6$, LKO $n = 5$, male) were measured after HFD feeding. The area under the curve (AUC) was used to quantify the GTT (J) and ITT (K) results. (L) The glycogen content in the livers of mice in (A) (WT $n = 8$, LKO $n = 6$, male). (M) Glucose infusion rate (GIR) during hyperinsulinemic–euglycemic clamp. GIR was calculated based on the 90 to 120 min glucose infusion (WT $n = 4$, LKO $n = 5$, male). (N) Suppression of hepatic glucose production (HGP) was calculated (WT $n = 4$, LKO $n = 5$, male). (O, P) Western blot analysis of essential markers of insulin signaling in the livers, white adipose tissues (WATs) and skeletal muscles from the WT and *Aspg* LKO mice. Before sacrifice, the mice were refed for 2 h after an overnight fasting (WT $n = 8$, LKO $n = 6$ for m; $n = 5$ each group for n, male). Data are represented as mean ± SEM. Statistical analysis was performed by unpaired two-tailed Student's *t* test for (C–G, J–N), by two-way ANOVA followed by Tukey's test for (H, I). Source data are available online for this figure.

transduction in the LKO mice livers did not cause any difference in the glucose tolerance test (GTT) (Fig. EV3J).

Next, we fed the male WT and *Aspg* LKO mice a high-fat diet (HFD) for 20 weeks. The *Aspg* LKO mice gained similar body weight as the control WT group (Fig. 3A). Although the hepatic appearance, weights, triglyceride contents and serum triglyceride, ALT and AST levels were comparable between the two groups (Fig. 3B–G), the diet-induced obese (DIO) LKO mice still showed decreased postprandial blood glucose levels while increased serum insulin levels (Fig. 3H,I). Moreover, HFD-fed LKO mice exhibited improved glucose tolerance and insulin tolerance (Fig. 3J,K). The enhanced glucose tolerance indicated a better capability to utilize and store glucose. The hepatic glycogenesis is a major direct effect of insulin on hepatic glucose metabolism (Petersen et al, 2017). As expected, the *Aspg* LKO mice had higher glycogen contents in the livers than the WT mice (Fig. 3L). Furthermore, a hyperinsulinemic–euglycemic clamp analysis was performed. The glucose infusion rate (GIR) required to maintain euglycemia was considerably higher, and hepatic glucose production (HGP) was more strongly suppressed by exogenous insulin in the *Aspg* LKO mice, indicating an improvement in the insulin resistance (Fig. 3M,N) To specifically delineate the tissues responsible for improved insulin sensitivity, western blot analysis was performed to detect the activation of canonical insulin pathway in the main insulin-responsive tissues. An augmentation in insulin-induced phosphorylation of PI3K (phosphoinositide-3-kinase), AKT (AKT serine/threonine kinase) and GSK3β (glycogen synthase kinase 3β) was observed in livers of *Aspg* LKO mice (Fig. 3O). Surprisingly, even white adipose tissues and skeletal muscles manifested more phosphorylation on AKT when hepatic *Aspg* was deleted (Fig. 3P). These phenotypes were not gender-specific since the rehabilitated glucose, insulin tolerance and hepatic insulin signaling were also observed in the female *Aspg* LKO mice (Fig. EV4A–E). Taken together, these data establish an important function of hepatic *Aspg* deficiency in improving systemic insulin signal transduction in DIO mice.

## ASPG upregulation in the liver exacerbates insulin signal transduction defects in obese mice

Then, we sought to determine whether ASPG upregulation would aggravate insulin signal transduction defects in obese mice. We intravenously injected AAV8-*Aspg* driven by the hepatocyte-specific thyroxine-binding globulin (TBG) promoter to both WT and LKO mice, and the AAV8-TBG-Vector viruses were used as

control. After the virus injection, the mice were fed a HFD for 16 weeks (Fig. EV5A). All the groups gained similar body weight during the course of HFD feeding (Fig. EV5B). When measured at the end of the experiment, the AAV8-TBG system specifically overexpressed ASPG by 2.2-fold in the livers of the WT mice or reconstituted ASPG levels in the LKO mice livers, without changing ASPG levels in white fat and skeletal muscles (Fig. EV5C). In contrast to the *Aspg* LKO mice, mice overexpressing *Aspg* in livers demonstrated the opposite phenotypes as visualized by more glucose intolerance, higher postprandial blood glucose levels and lower insulin levels than the control group (WT + AAV8-ASPG versus WT + AAV8-VEC) (Fig. 4A–C). AAV-mediated *Aspg* restoration in the LKO mice livers abolished all the metabolic benefits accomplished by the deficiency of *Aspg*, including improved glucose tolerance, hypoglycemia and high insulin levels after meals (Fig. 4A–C).

Given that the postprandial serum insulin levels negatively correlate with the hepatic ASPG expression (Fig. 4C), we reasoned that ASPG in the liver might inhibit insulin secretion in some way. To verify the hypothesis, we executed a glucose-stimulated insulin secretion (GSIS) assay. As shown in Fig. 4D, HFD for 16 weeks dramatically ruined the capability of pancreas β cells to secrete insulin upon glucose stimulus (WT + HFD + AAV8-VEC versus WT+Chow+AAV8-VEC), consistent with the previous study that HFD feeding for long duration elicited pancreas islet functional decompensation (Wang et al, 2022). Overexpressing *Aspg* further worsened, but ablating *Aspg* salvaged decreased insulin secretion (Fig. 4D). Immunofluorescence staining revealed that the mice with overexpressed ASPG had less pancreas β cell masses. Deleting *Aspg* preserved the pancreas islet after a long-term HFD feeding, and the protective effect was extinguished by the reconstitution of *Aspg* in the LKO mice (Fig. 4E,F). Moreover, the *Aspg* level was also negatively associated with HOMA-β, the index of insulin secretion function, in patients with MASLD (Fig. 4G).

To further decipher the effects of hepatic ASPG on systemic insulin signal transduction, we evaluated the activation of the insulin signaling pathway by western blot in the *Aspg*-over-expressed mice fed on a HFD. A high level of ASPG expression in liver did not influence the liver mass, levels of fasting blood glucose and serum insulin, ALT and AST (Fig. EV5D–G), but profoundly exacerbated defects in systemic insulin signal transduction, as evidenced by less phosphorylation of the major nodes of insulin signaling pathway in the livers, white adipose tissues and skeletal muscles (Fig. 4H,I). The decreased hepatic glycogen contents as indicated by periodic acid-schiff (PAS) staining further

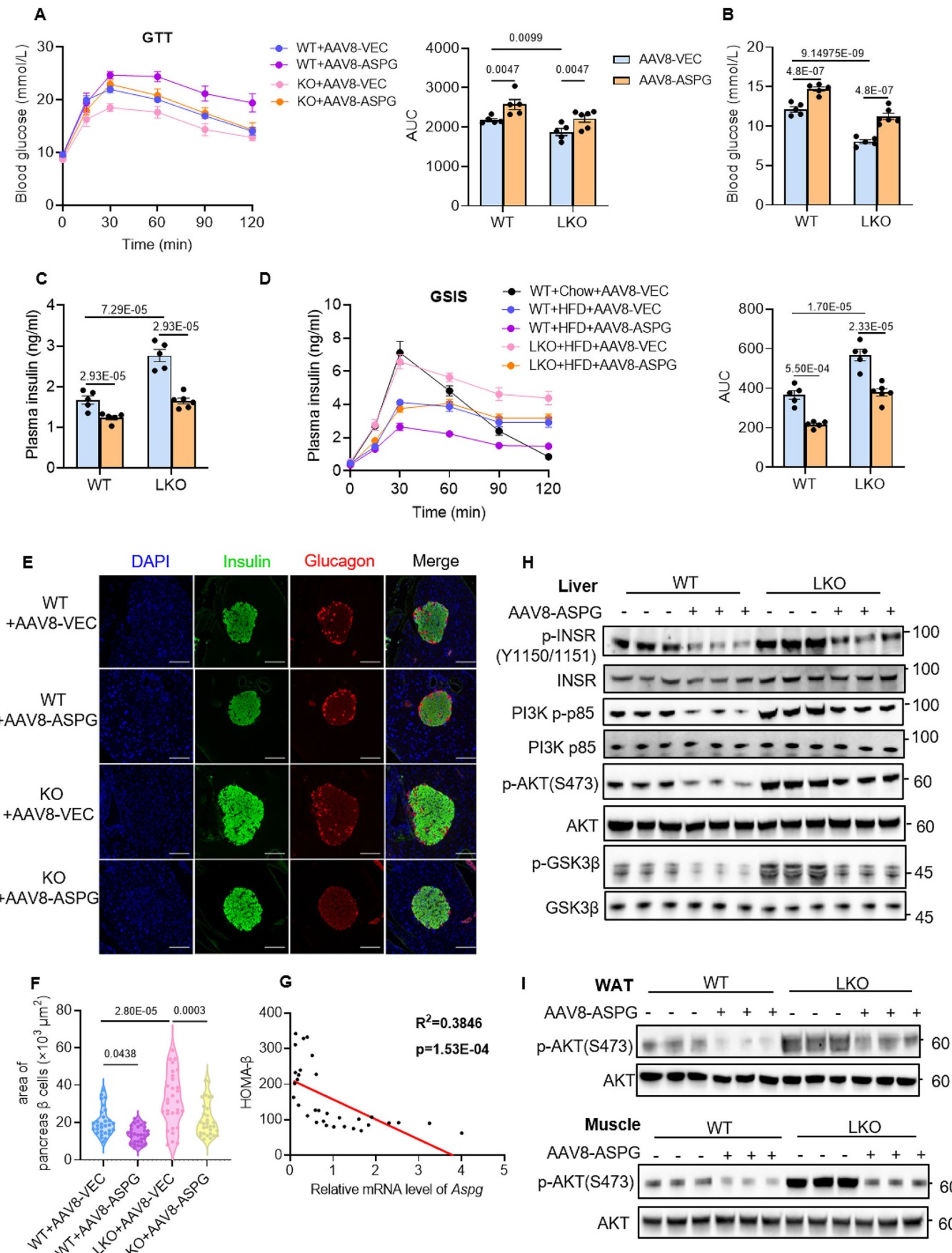

**Figure 4. Hepatic overexpression of ASPG exacerbates HFD-induced defects in insulin signal transduction.**

(A–D) AAV-TBG-*Aspg* or AAV-TBG-Vector was expressed in the female WT and *Aspg* LKO mice livers, followed by a HFD feeding for 16 weeks. GTT (A), the postprandial blood glucose (B) and plasma insulin (C) levels, and glucose-stimulated insulin secretion (GSIS) (D) were measured. WT + AAV8-VEC $n = 5$, WT + AAV8-ASPG $n = 5$, LKO + AAV8-VEC $n = 5$, LKO + AAV8-ASPG $n = 6$, WT+Chow+AAV8-VEC $n = 5$; female mice. (E) Representative immunofluorescence staining of insulin (green) and glucagon (red) in pancreases. Nuclei were stained with DAPI (blue) (scale bar 50 μm). (F) Quantification of the pancreas β-cell area in the immunofluorescence slices in (E). $n = 30$ pancreas islets in each group. (G) The correlation between *Aspg* mRNA levels and HOMA-β in patients with MASLD ($n = 32$). (H, I) Western blot analysis of essential markers of insulin signaling in the livers, WATs and muscle tissues. Mice were sacrificed at 2 h of fasting-refeeding transition ($n = 3$ each group). Data are represented as mean ± SEM. Statistical analysis was performed by two-way ANOVA followed by Tukey's test for (A–D), by one-way ANOVA followed by Tukey's test for (F). The Pearson and Spearman correlation analysis was applied for (G). Source data are available online for this figure.

confirmed the aggravation of hepatic insulin resistance upon upregulating ASPG expression in livers (Fig. EV5H,I). Strikingly, restoration of the *Aspg* level in the knockout mice livers entirely wiped out the enhanced insulin signaling transduction (Fig. 4H,I). Fat mass, lean mass, hepatic inflammation and fibrosis also play important roles in regulating systemic insulin sensitivity (Merz and Thurmond, 2020; James et al, 2021; Miao et al, 2024). Herein, overexpressing or knocking-out ASPG in livers exerted no influence on mice fat mass, lean mass, or marker genes expression of hepatic inflammation and fibrosis (Fig. EV5J–M). Taken together, these data strongly suggested although it was of minor importance for the progression of hepatic steatosis, a high level of hepatic ASPG exacerbated impairment of systemic insulin signal transduction.

## *Aspg* deficiency improves glucose tolerance in the late T2D mouse model

Glucose homeostasis is regulated by insulin and the responsiveness of target tissues to insulin stimulation. Considering the altered postprandial serum insulin levels in *Aspg* LKO mice, we queried whether the ameliorated glucose tolerance was entirely due to the better competency of insulin secretion. To answer this question, we established a late stage of T2D mouse model by repeated administration of low doses of streptozotocin (STZ) to the HFD-fed WT and *Aspg* LKO mice (Fig. EV6A) (Furman, 2021). STZ selectively destroys pancreatic islet β cells to cause insulin deficiency and hyperglycemia. The high blood glucose and low fed insulin levels indicated a diabetes model had been set up, and simultaneously STZ eliminated the difference in serum insulin levels (after meal) between the WT and LKO mice (Fig. EV6B,C). In such circumstances, the *Aspg* LKO mice still showed lower levels of random blood glucose, improved glucose tolerance and systemically intensified insulin signaling (Fig. EV6D–G). Similar to the previous observations, the liver mass, serum ALT and AST levels were comparable between the WT and LKO mice (Fig. EV6H,I). Thus, we concluded that the enhanced insulin sensitivity in peripheral tissues as well as a better pancreas islet function contributed to blood glucose homeostasis in the *Aspg* LKO mice.

## Increased LPIs in *Aspg* knockout hepatocytes enhance insulin signaling transduction

To explore the molecular mechanism how the paucity of ASPG inspired insulin signaling transduction, we first isolated primary hepatocytes from the WT and *Aspg* LKO mice to check their responsiveness to insulin. The proximal insulin signaling axis, including INSR (insulin receptor), PI3K and AKT, showed robust phosphorylation in the LKO hepatocytes upon insulin stimulation (Fig. 5A). On the contrary, *Aspg* overexpression by adenovirus infection blunted insulin signaling in primary hepatocytes (Fig. 5B). These data implied ASPG inhibited hepatic insulin signaling in a cell-autonomous manner.

Considering that ASPG bears lysophospholipase activity, we deduced a different lipid composition might present in the LKO mice livers. Therefore, the liver tissues from the HFD-fed WT and LKO mice were subjected to a lipidomics assay. The data demonstrated that lipid species that were well-known to cause insulin resistance, such as ceramide and diacylglycerol (DAG), were comparable between the WT and LKO mice (Fig. EV7A). The total amounts of triacylglycerol (TAG), phospholipids, cholesterol ester (chE) and sphingomyelin (SM) showed no difference either (Fig. EV7A). Strikingly, there were considerable increases in the quantity of total LPIs and subspecies of LPI with diverse chain length and degree of unsaturated fatty acids in the LKO mice livers (Fig. 5C,D). However, the serum LPI levels were similar between genotypes (Fig. EV7B). Quantitative analysis of intracellular and extracellular LPI levels was also performed in primary hepatocytes. Not surprisingly, there was a higher level of LPI inside the hepatocytes deficient of *Aspg*. The extracellular LPI level showed no difference between the WT and *Aspg* knockout hepatocytes (Fig. EV7C,D). Using the decrease in the amount of lysophospholipid substrates as a readout of ASPG catalytic activity, we confirmed that the purified recombinant mouse ASPG protein could hydrolyze LPI 18:0 in vitro, but it showed no activity towards LPC 18:0 and LPE 18:0 (Fig. 5E).

Bioactive lipid LPIs are identified as agonists for GPR55, and their biological activities are divided as GPR55-mediated and non-GPR55-mediated effects (Arifin and Falasca, 2016). Increased LPIs levels did not alter the expression of *Gpr55* and lysophosphatidy-linositol acyltransferase 1 *Mboat7* in the *Aspg* LKO livers (Fig. EV7E). In accord with the previous study that LPI treatment could enhance insulin-dependent AKT phosphorylation in HepG2 cells (Lipina et al, 2019), we showed LPI dramatically reinforce insulin signaling in primary hepatocytes in an insulin-resistant state induced by palmitate treatment (Fig. 5F).

In order to unravel the role of GPR55 in the LPI-enhanced insulin signaling, we treated the primary hepatocytes with GPR55 antagonist ML-193. ML-193 blunted LPI-reinforced insulin signal transduction in the hepatocytes (Fig. 5F). Unexpectedly, the knockout cells still showed an enhanced insulin signaling but to a less degree after GPR55 was blocked by ML-193 (Fig. 5G), which indicated LPI promoted insulin sensitivity through both GPR55-

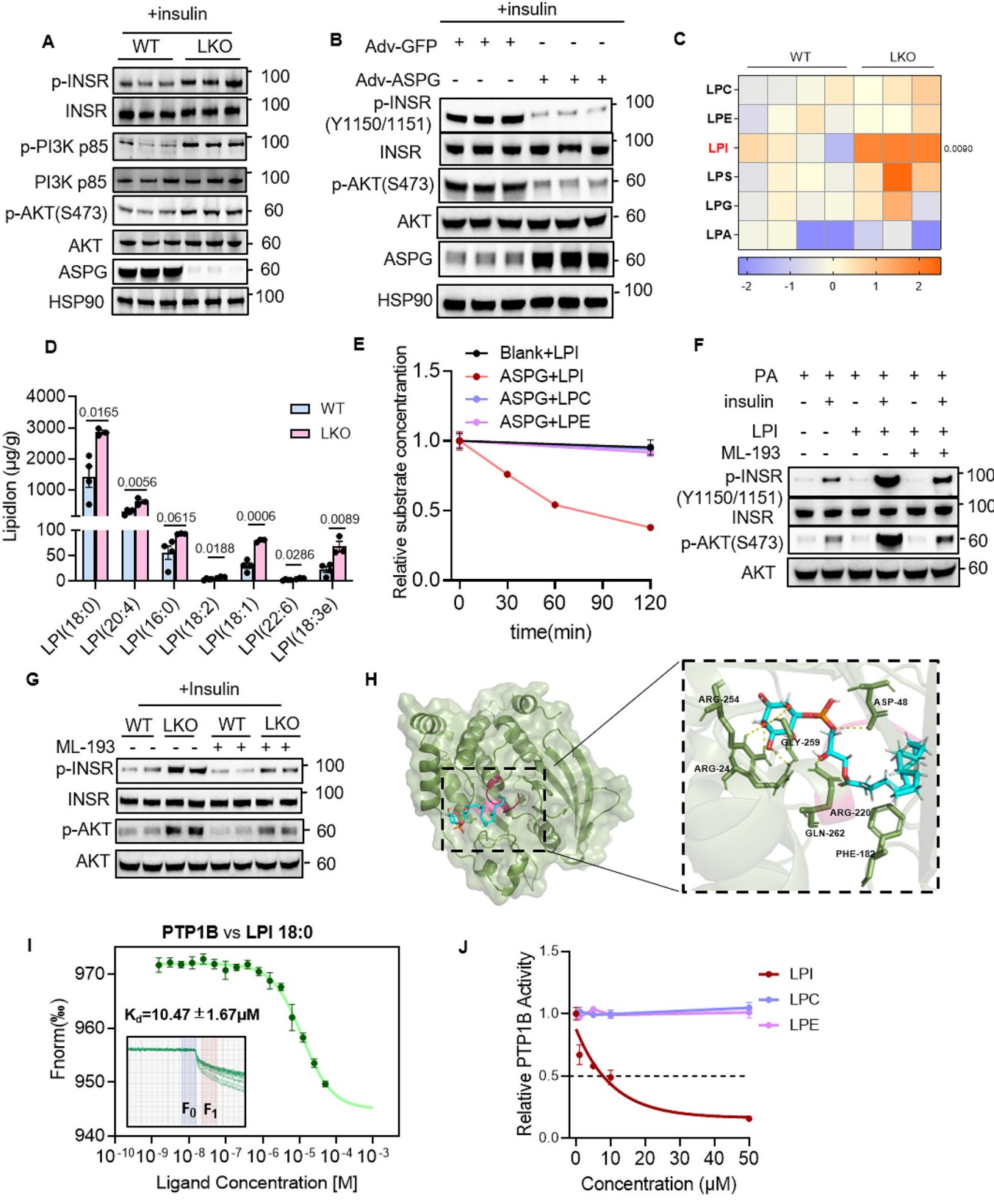

**Figure 5. Increased LPIs in *Aspg* knockout hepatocytes enhance insulin signaling via suppressing PTP1B activities.**

(A) Western blot analysis of the indicated proteins in the primary hepatocytes isolated from the WT and *Aspg* LKO mice. Insulin (10 nM) stimulated the cells for 15 min before harvesting the cells ($n = 3$ each group). (B) Adenoviruses coding for *Aspg* (Adv-ASPG) or vector control (Adv-GFP) infected the primary hepatocytes. The cells were stimulated by insulin (10 nM) for 15 min before the western blot assay ($n = 3$ each group). (C, D) Lipidomic analysis for the liver tissues from the WT and *Aspg* LKO mice fed a HFD for 20 weeks. A heatmap of the total amount of each species of lysophospholipid was presented in (C). The quantification of the major LPI subspecies was shown in (D). WT $n = 4$, LKO $n = 3$; female. (E) The ASPG lysophospholipidase activity assay in vitro. Purified ASPG and lysophospholipid substrates were incubated at 37 °C for the indicated time. The decrease in the substrate levels was subsequently quantified by UHPLC–MS/MS ($n = 6$ each time point). (F) Primary hepatocytes isolated from female C57BL/6 mice were subjected to treatment as indicated followed by western blot analysis. (G) Primary hepatocytes isolated from the female WT and *Aspg* LKO mice were treated with or without ML-193 (10 μM) for 6 h. Western blot analysis was performed after 15-min insulin (10 nM) stimulation. $n = 2$ in each group. (H) Interactive visualization of LPI 18:0 binding to N-terminus of human PTP1B. The P-loop of PTP1B is shown in red. (I) The binding affinity between PTP1B and LPI 18:0 was measured by microscale thermophoresis (MST). Inset, thermophoretic movement of fluorescently labeled proteins. $K_d$, dissociation constant. $n = 4$ independent measurements. (J) The in vitro PTP1B activity assay in the presence of LPI 18:0, LPC 18:0 or LPE 18:0. The PTP1B activity without any lysophospholipids was marked as 1 ($n = 3$ each group). Data are represented as mean ± SEM. Statistical analysis was performed by unpaired two-tailed Student's $t$ test for (C, D). For (F, G), the experiments were repeated at least for three times and representative images were shown. Source data are available online for this figure.

mediated and non-GPR55-mediated ways. Considering that ASPG and LPI affect insulin signaling cascade at the beginning of INSR, and PTP1B is the main negative regulator to dephosphorylate INSR (Teimouri et al, 2022), we surmised intracellular LPI might inhibit PTP1B activity to boost INSR-mediated insulin signaling transduction. There are 4 conserved loop regions in PTP1B: P- (His214-Arg221), WPD (Thr177-Pro188), Q- (Gly259-Thr263) and pTyr recognition (Tyr46-Val49) loops (Cui et al, 2019). MCULE 1-click-docking website (https://mcule.com/apps/1-click-docking/) predicted that the acyl chain of LPI 18:0 occupied the catalytic P-loop of human PTP1B by forming van der waals forces with Ser216, Ala217, Ile219, Gly220 and Arg221 residues. All these amino acids were conserved in human and mouse PTP1B (Figs. 5H and EV7F,G). Furthermore, microscale thermophoresis (MST) measurement showed LPI 18:0 and PTP1B dissociation constant ($K_d$) of 10.47 ± 1.67 μM (Fig. 5I). In vitro PTP1B inhibitor screening assay revealed 10 μM LPI 18:0 inhibited PTP1B activity by 50%, and the suppressive effect was LPI dose-dependent. As a control, LPC 18:0 and LPE 18:0 could not suppress PTP1B activity at all (Fig. 5J). Knocking down PTP1B largely increased INSR and AKT phosphorylation. However, loss of PTP1B could not further enhance insulin signaling in the *Aspg* knockout or LPI-treated hepatocytes (Fig. EV7H,I), indicating the importance of PTP1B in LPI-mediated insulin-sensitizing effects.

## Ablation of *Aspg* reduces hepatokine SELENOP expression

ASPG deficiency only increased the intracellular LPI level and did not change serum LPI concentration. However, the hepatic ASPG-LPI axis exerted influence on pancreatic insulin secretion and insulin signal transduction in adipose tissues and skeletal muscles, which implied other factors downstream of the ASPG-LPI axis were involved in the systemic metabolic regulation. Hepatokines, that are proteins secreted by hepatocytes, have been linked to insulin resistance in non-hepatic tissues during the progression of hepatic steatosis (Meex and Watt, 2017). To probe whether and which hepatokine participated in the ASPG-associated insulin resistance, we measured several hepatokines' mRNA levels by quantitative PCR in the WT and LKO livers, and especially paid attention to the hepatokines associated with insulin secretion and insulin resistance (Meex and Watt, 2017; Watt et al, 2019). The hepatic mRNA levels of *Sepp1* and serum SELENOP protein levels were significantly

decreased in both male and female HFD-fed LKO mice (Figs. 6A and EV8A), and were further confirmed in STZ/HFD as well as in chow diet-fed *Aspg* LKO mice (Fig. 6B,C). In contrast, overexpression of *Aspg* in livers increases serum SELENOP contents. AAV-mediated *Aspg* restoration in the LKO mice slightly but significantly raised SELENOP levels as compared to the LKO mice (Fig. 6D). SELENOP is a well-known hepatokine to cause insulin resistance, which reduces phosphorylated AKT in liver and muscle, and decreases the number of pancreas β cells and GSIS (Misu et al, 2010; Mita et al, 2017). A positive correlation between the hepatic *Aspg* and *Sepp1* expression levels was also observed in human liver specimens (Fig. 6E).

Then we isolated primary hepatocytes and expressed *Aspg* by adenovirus infection. High levels of *Aspg* drove SELENOP expression and secretion, and palmitic acid treatment further augmented SELENOP production (Fig. EV8B,C). When the 3T3L1 adipocytes were subjected to the conditional media incubation that were collected from primary hepatocytes overexpressing *Aspg*, decreases in the phosphorylated INSR and AKT were perceived (Fig. EV8D).

It was reported that FOXO1 upregulated and insulin down-regulated transcription of *Sepp1* by phosphorylating and inactivating FOXO1 (Speckmann et al, 2008; Walter et al, 2008). We diminished *Sepp1* expression and secretion in primary hepatocytes by knocking down *Foxo1*, confirming the weight of FOXO1 in regulating *Sepp1* level (Fig. EV8E,F). Indeed, AAV-mediated ASPG overexpression in mouse liver dramatically reduced FOXO1 phosphorylation (Fig. 6F). On the contrary, increased phosphorylated FOXO1 appeared in the LKO mice livers (Fig. 6F), which should be responsible for the decreased *Sepp1* expression in ASPG-deficient hepatocytes. Interfering with the insulin signaling pathway by *Ptp1b* siRNA or an AKT inhibitor (AKTi) also abolished the difference in the *Sepp1* levels between the primary WT and LKO hepatocytes (Figs. 6G and EV8G). Moreover, LPI synergized with insulin to decrease the expression of *Sepp1* in hepatocytes, and AKTi eliminated this effect (Fig. EV8H). Considering that ML-193 treatment did not alter *Sepp1* mRNA and protein levels in either WT or LKO hepatocytes, we would think the ASPG-LPI axis regulated *Sepp1* expression independent of GPR55 (Fig. EV8I,J). Collectively, these results indicated that hepatocyte-specific *Aspg* knockout could enhance insulin signal transduction; subsequently, increased phosphorylated FOXO1 reduced *Sepp1* expression, leading to whole-body metabolic protective effects.

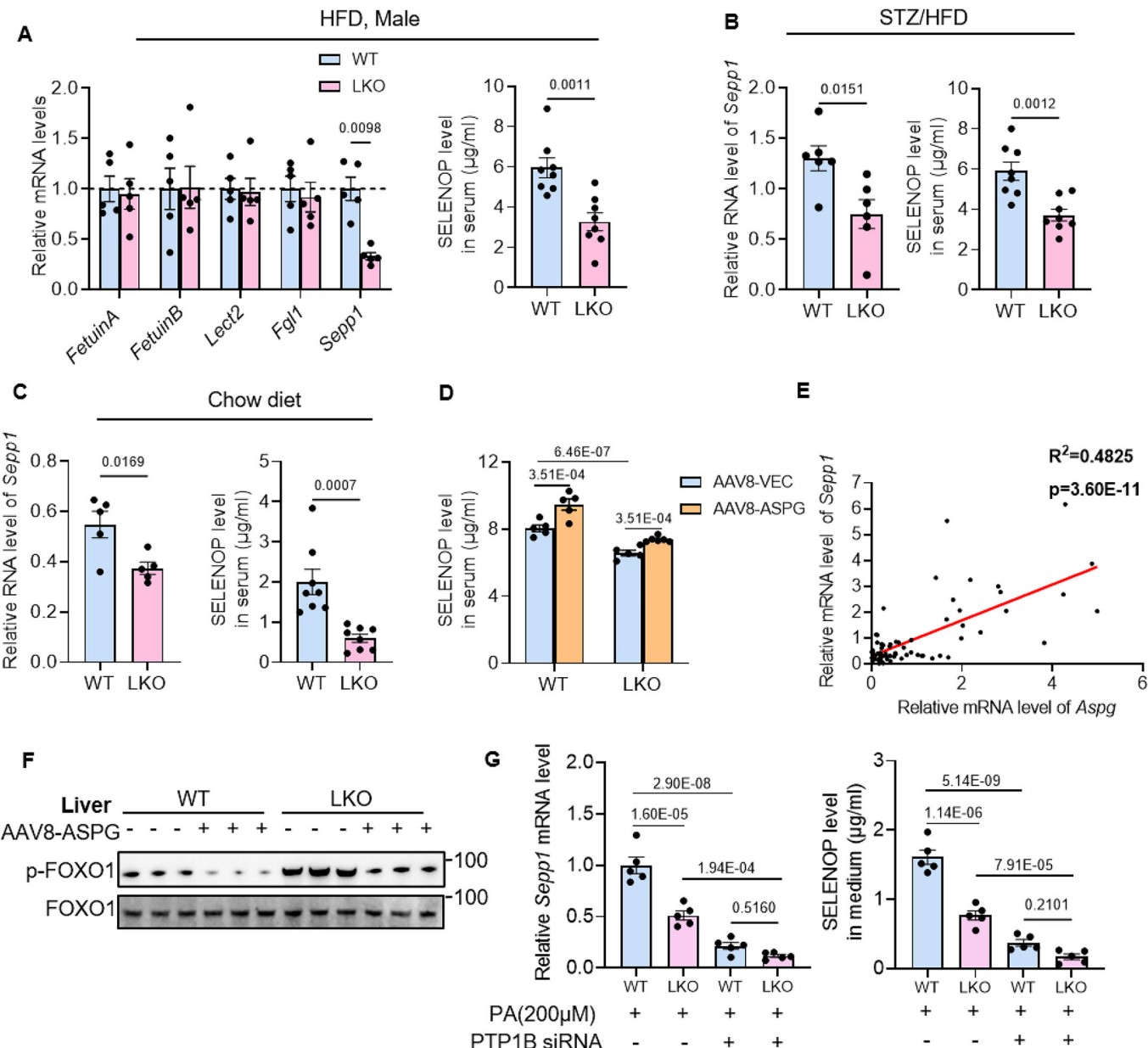

**Figure 6. Loss of ASPG inhibits hepatokine SELENOP expression.**

(A) Relative mRNA levels of the selected hepatokines in livers (left, $n = 5$, male), and SELENOP protein levels in serum (right, $n = 8$, male) of the WT and *Aspg* LKO mice fed a HFD for 20 weeks. (B) Relative mRNA levels of *Sepp1* in livers (left, $n = 6$, male), and SELENOP protein levels in serum (right, $n = 8$, male) of the STZ/HFD WT and *Aspg* LKO mice. (C) Relative mRNA levels of *Sepp1* in livers (left, $n = 5$, male), and SELENOP protein levels in serum (right, $n = 8$, male) of the chow diet-fed WT and *Aspg* LKO mice. (D) Serum SELENOP protein levels in mice expressing AAV-ASPG as described in Fig. 4A (WT + AAV8-VEC $n = 5$, WT + AAV8-ASPG $n = 5$, LKO + AAV8-VEC $n = 5$, LKO + AAV8-ASPG $n = 6$; female). (E) The correlation between *Aspg* and *Sepp1* mRNA levels in human liver specimens ($n = 69$). (F) Western blot analysis of the level of FOXO1 in the liver tissues from the mice expressing AAV-ASPG as described in Fig. 4A ($n = 3$ each group). (G) Relative mRNA levels of *Sepp1* (left), and secreted SELENOP levels (right) in primary hepatocytes isolated from the WT and LKO mice. The cells were transfected with or without *Ptp1b* siRNA and were treated with PA for 24 h. $n = 5$ each group. Data are represented as mean ± SEM. Statistical analysis was performed by unpaired two-tailed Student's *t* test for (A–C), by two-way ANOVA followed by Tukey's test for (D), by Pearson and Spearman correlation analysis for (E), by one-way ANOVA followed by Tukey's test for (G). Source data are available online for this figure.

## Reducing SELENOP expression recovers glucose and insulin tolerance in ASPG-overexpression mice

In order to assess the role of SELENOP in the ASPG-led insulin signaling defects, we tried to inhibit SELENOP expression by infusing AAV-*Ptp1b* shRNAs or AAV-*Sepp1* shRNAs to *Aspg*-overexpressing mice. After 8-week HFD feeding, *Aspg*-overexpressing mice showed a considerable increase in the pyruvate tolerance test (PTT), indicating a compromised insulin signaling (Fig. EV9A). Then, AAV-shRNAs targeting *Ptp1b* or *Sepp1* were infused to these

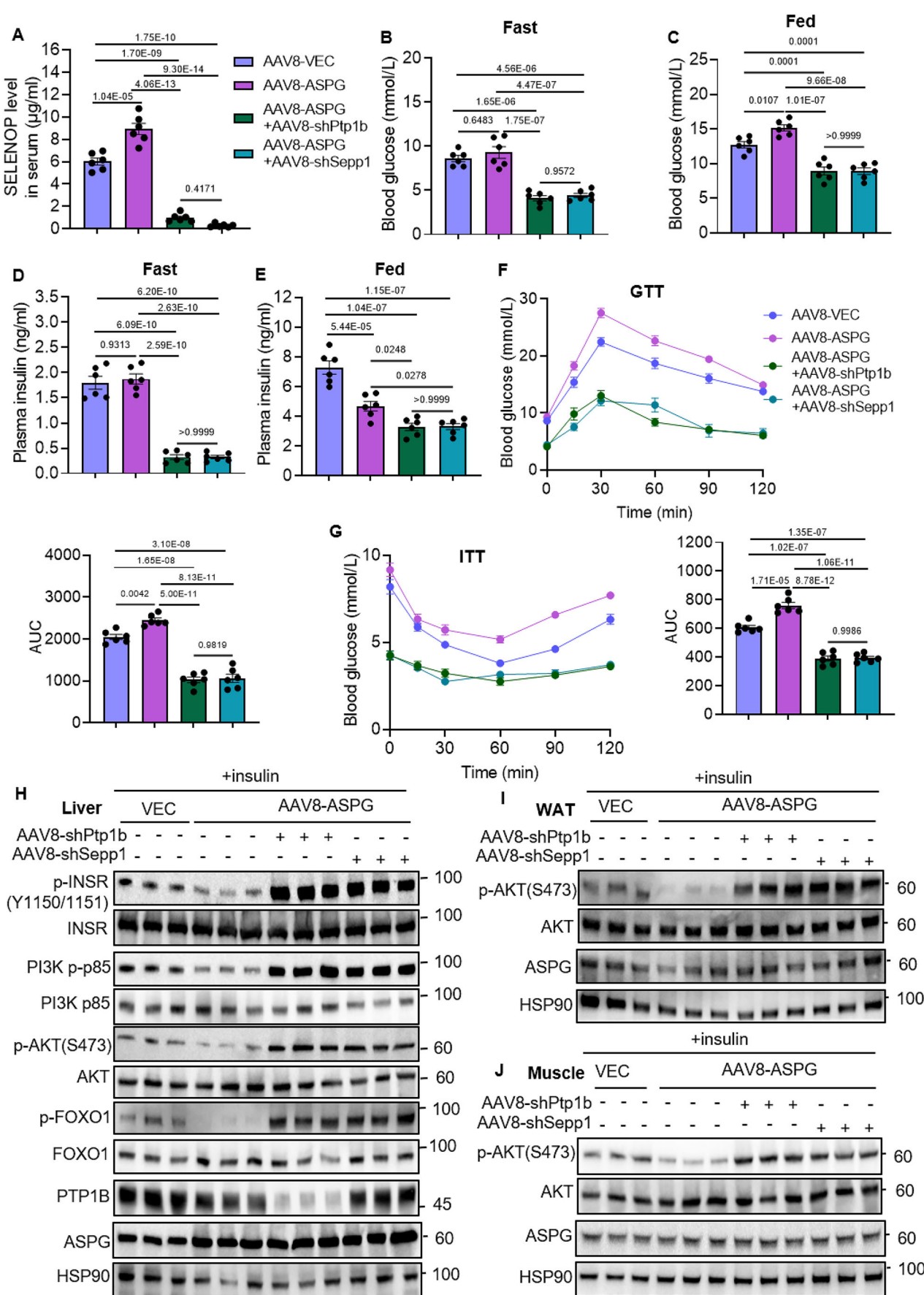

◄

**Figure 7.  *Ptp1b* or *Sepp1* knockdown restores insulin signaling in ASPG-overexpression mice.**

(A) Serum SELENOP protein levels were measured in mice as indicated. AAV-ASPG and AAV-shRNAs were infused to C57BL/6 mice as described in Fig. EV9B. $n = 6$ each group, male. (B–E) The fasting (B) and postprandial blood glucose (C), fasting (D), and postprandial plasma insulin (E) levels in mice of (A). $n = 6$, each group, male. (F, G) GTT and ITT assays were performed for mice in (A). AUC was used to quantify the GTT (G) and ITT. $n = 6$, each genotype, male. (H–J) Western blot analysis of the insulin signaling pathway in the livers (H), WATs (I), and muscle tissues (J). Insulin was injected to the 16h-fasted mice 15 min before the assay ($n = 3$ each group, male). Data are represented as mean ± SEM. Statistical analysis was performed by one-way ANOVA followed by Tukey's test for (A–G). Source data are available online for this figure.

*Aspg*-overexpression mice (Fig. EV9B). The serum SELENOP levels, fasting or postprandial glucose and insulin levels were all dramatically reduced in both *Ptp1b* and *Sepp1* knockdown groups (Fig. 7A–E). Deficiency of *Ptp1b* or *Sepp1* in hepatocytes also improved glucose and insulin tolerance as indicated by GTT and ITT assay (Fig. 7F,G). Simultaneously, during the GTT assay, serum insulin levels were analyzed at 0, 15, and 30 min. The *Aspg*-overexpression mice had declined GSIS ability (Fig. EV9C), possibly owing to ASPG accelerating islet β cell loss as previously shown in Fig. 4E. Intriguingly, mice with *Ptp1b* or *Sepp1* knockdown secreted much less insulin upon glucose stimulus than the *Aspg*-overexpression mice (Fig. EV9C). However, they always kept low blood glucose levels during the GTT assay (Fig. 7F), suggesting that a restored systemic insulin sensitivity abrogated the requirement of high insulin secretion. Indeed, liver, adipose tissues and skeletal muscle manifested enhanced insulin signaling transduction upon insulin injection in both *Ptp1b* and *Sepp1* knockdown mice (Fig. 7H–J). Together, these in vivo experiments further validated that PTP1B and SELENOP played an important role in the ASPG-elicited insulin transduction defects.

## Targeting hepatic ASPG ameliorates insulin signaling defects in MASLD mice

MASLD is the risk factor for insulin resistance and T2D. The progression of T2D is accompanied by an average of 30% loss in β-cell mass (McCarty et al, 2024). As such, we were interested in evaluating the therapeutic efficacy of targeting ASPG to ameliorate steatosis-associated systemic insulin signal defects. We utilized AAV-TBG-Cre to diminish hepatic *Aspg* in *Aspg*^flox/flox mice that had already been fed a HFD for 10 weeks. After virus injection, mice were fed a HFD for another 6 weeks (Fig. 8A). AAV-TBG-Cre virus efficiently abolished *Aspg* expression in livers but not in adipose tissues and skeletal muscles (Fig. 8B). Loss of hepatic ASPG dramatically alleviated glucose intolerance, potentiated insulin secretion in response to glucose feeding, and therefore manifested a low postprandial blood glucose level and a high insulin level (Fig. 8C–F). Although the absence of ASPG in the liver did not affect the body weight, liver mass, serum ALT and AST levels (Fig. 8G–I), a robust enhancement of insulin signaling transduction was observed in the livers, white fat tissues and muscles in reply to feeding-induced insulin secretion (Fig. 8J–L).

## Discussion

Previously, ASPG is named 60-kDa lysophospholipase and its functional role especially in vivo is poorly characterized. Here, we discerned insulin signaling pathway transcriptionally

downregulated *Aspg* expression. The low level of ASPG facilitates hepatic insulin sensitivity by boosting intracellular LPI levels to inhibit PTP1B activity, and further decreases SELENOP expression which forms a positive feedback loop to reinforce systemic insulin signal transduction. A strong correlation between hepatic ASPG level and insulin resistance index was observed in MASLD patients. In the diet-induced steatosis mouse model, inhibiting hepatic ASPG expression improved glucose tolerance and systemic insulin signaling transduction. Our work highlights the particular function of intracellular LPI to suppress PTP1B activity, and targeting ASPG-LPI axis might be a potential strategy for strengthening insulin signal transduction.

The lysophospholipase activity of ASPG was determined previously using lysates from ASPG-overexpressing cells (Sugimoto et al, 1998; Menniti et al, 2010). In our study, lipidomics analysis of the liver tissues uncovered levels of total and major species of LPI were significantly increased in the *Aspg* knockout mice livers, substantiating the lysophospholipase activity of ASPG in vivo. We further purified the ASPG protein and confirmed its activity towards LPI as substrates by LC-MS quantification. The biological effects caused by LPI accumulation due to *Aspg* ablation were different from those observed in the *Mboat7* (membrane-bound enzyme O-acyltransferase domain-containing 7) knockout mice. MBOAT7, also known as lysophosphatidylinositol acyltransferase 1, selectively incorporates polyunsaturated fatty acids (PUFAs) into LPI at the sn-2 position (Lee et al, 2012; Caddeo et al, 2021). The *Mboat7* loss-of-function variant rs641738TT is associated with severe liver diseases in human, which is likely to occur through phosphatidylinositol (PI) remodeling (Varadharajan et al, 2022). Studies in mice also found that *Mboat7* ablation could promote hepatic steatosis, inflammation and fibrosis (Helsley et al, 2019; Thangapandi et al, 2021). A high rate of PI turnover, increased total LPI level but a decreased level of PI 38:4 might account for the pathologies in the *Mboat7*-depleted livers (Tanaka et al, 2021; Thangapandi et al, 2021). In addition, deficiency of *Mboat7* in liver had a profound impact on the levels of other lipid species, including phosphatidylglycerol (PG), lysophosphatidylglycerol (LPG) and phosphatidic acid (PA) (Thangapandi et al, 2021). The different lipidomic profile might explain the phenotypic diversities between *Aspg*- and *Mboat7*-deficient mice.

Extracellular LPI induces hepatic steatosis and directly activates the hepatic stellate cells (HSCs) in a GPR55-dependent manner (Fondevila et al, 2021). Here, we find that deletion of *Aspg* mainly caused an increase in LPI level inside hepatocytes (see Fig. EV7C,D). Intracellular LPIs inhibit PTP1B activities by binding to its catalytic pocket (P-loop). PTP1B is the predominant INSR and insulin receptor substrate (IRS) tyrosine phosphatase that negatively regulates insulin signaling, and has been a promising therapeutic target for treating T2D. Cellular bioavailability and

permeability are the two major hindrances in the development of clinical drugs targeting PTP1B (Delibegović et al, 2024). The intracellular accumulation of LPI caused by loss of *Aspg* overcomes these obstacles, which represents a novel strategy to suppress PTP1B and increase insulin sensitivity.

The impacts of ASPG-LPI axis on MASH and hepatic fibrosis have not been tested yet. We speculate that the amelioration of hepatic insulin sensitivity would partly protect from overnutrition-associated chronic inflammation and fibrosis in the livers of *Aspg* knockout mice. It is reported that supplementation of LPI activates collagen synthesis in HSCs in a GPR55-dependent manner, which emphasizes the role of extracellular LPIs as ligands for GPR55 (Fondevila et al, 2021). In their study, a methionine/choline-deficient diet-induced or CCl$_4$-induced steatohepatitis model was adopted, which did not involve insulin resistance in the process of disease. Thus, it is valuable to test the effects of the intracellular LPI accumulation caused by *Aspg* deficiency on the progression of MASH and hepatic fibrosis on the grounds of improved insulin sensitivity.

In conclusion, this study unveils a unique role of the hepatic ASPG-LPI-SELENOP axis in the regulation of systemic insulin signal transduction. The suppression of PTP1B by intracellular LPIs represents an unexpected lysophospholipid-mediated signaling pathway independent of their receptors. The decreased expression of hepatokine SELENOP in the *Aspg* knockout mice brings out beneficial metabolic effects in the pancreas, muscle and adipose tissues. These insights propose a pharmacological inhibitor of ASPG as a potential therapeutic avenue for the improvement of dysfunctional insulin signal transduction.

# Methods

### Reagents and tools table

| Reagent/resource | Reference or source | Identifier or catalog number |
|---|---|---|
| **Experimental models** | | |
| Aspg-Floxed mice | GemPharmatech | T022400 |
| Alb-Cre mice | GemPharmatech | T003814 |
| db/db | GemPharmatech | T001461 |
| Aspg LKO mice | This study | N/A |
| **Recombinant DNA** | | |
| PGL3-hAspg-WT promoter | This study | N/A |
| PGL3-hAspg-Mut promoter | This study | N/A |
| PGL3-mAspg-WT promoter | This study | N/A |
| PGL3-mAspg-Mut promoter | This study | N/A |
| PET28a-His-Aspg | This study | N/A |
| PET28a-His-Ptp1b | This study | N/A |
| **Antibodies** | | |
| ASPG | Sigma-Aldrich | #SAB1301923 |
| HSP90 | Proteintech | #13171-1-AP |

| Reagent/resource | Reference or source | Identifier or catalog number |
|---|---|---|
| β-tubulin | Proteintech | #13171-1-AP |
| CANX | Cell Signaling Technology | # 2146 |
| LMNA | Proteintech | #10427-2-AP |
| β-catenin | Proteintech | # 10298-1-AP |
| VDAC1 | Proteintech | # 51067-2-AP |
| p-INSR(Tyr1150/1151) | Cell Signaling Technology | #3024 |
| INSR | Cell Signaling Technology | #3025 |
| p-AKT (Ser473) | Cell Signaling Technology | #4060 |
| AKT | Cell Signaling Technology | #4685 |
| p-FOXO1 | Cell Signaling Technology | #2599 |
| FOXO1 | Cell Signaling Technology | #2880 |
| PCK1 | Cell Signaling Technology | #12940 |
| PI3K-p-p85 | Abclonal | #AP0854 |
| PI3K-p85 | Abclonal | #A4992 |
| p-GSK3β | Cell Signaling Technology | #9322 |
| GSK3β | Cell Signaling Technology | #12456 |
| PTP1B | Proteintech | #11334-1-AP |
| HRP, anti-rabbit IgG | Proteintech | #SA00001-2 |
| **Oligonucleotides and other sequence-based reagents** | | |
| qRT-PCR primers | This study | Appendix Table S3 |
| shRNA and siRNA target | This study | Appendix Table S4 |
| ChIP-qPCR primers | This study | Appendix Table S5 |
| **Chemicals, enzymes, and other reagents** | | |
| Collagenase IV | Sigma-Aldrich | #C5138 |
| D-(+)-Glucose (≥99%) | Vetec | #V900392 |
| DMEM | MeilunBio | #MA0212 |
| Fetal Bovine Serum | Hyclone | #SH30070.03 |
| Newborn Calf Serum | Hyclone | #SH30073.04 |
| PMSF | YEASEN | #20104ES03 |
| Protease inhibitor cocktail | YEASEN | #20109ES05 |
| High-fat-diet | Research Diet | #D12492 |
| Penicillin–Streptomycin | Hyclone | #SV30010 |
| Trypsin-EDTA | Gibco | #25200056 |
| Protein A-agarose beads | Cytiva | #17078001 |
| Ni-beads | Smart lifesciences | #A20091701 |
| Sodium palmitate | Sigma-Aldrich | #P0500 |
| Bovine serum albumin | Sigma-Aldrich | #9048-46-8 |
| Insulin | Sigma-Aldrich | #I3536 |
| Trizol | Vazyme | #R401-01 |
| SYBR Green Supermix | Vazyme | #Q511-02/03 |
| 18:0 LPI | Avanti Polar Lipids | #850091P |
| 18:0 LPE | Avanti Polar Lipids | #855775P |

| Reagent/resource | Reference or source | Identifier or catalog number |
|---|---|---|
| 18:0 LPC | Avanti Polar Lipids | #856715P |
| ML-193 | MCE | #HY-110125 |
| Akt inhibitor | MCE | # HY-10355 |
| DiOC6(3) | Shanghai MaoKang Biotechnoloy | #MX4009 |
| **Software** | | |
| Image J | National Institutes of Health (NIH) | RRID: SCR_003070 |
| GraphPad Prism 9 | GraphPad software | RRID: SCR_002798 |
| **Other** | | |
| N/A | N/A | N/A |

## Human liver tissues

All procedures involving human specimens in this study complied with the ethical guidelines of the Declaration of Helsinki and Istanbul. Ethical approval was provided by the Ethics Committee Board at Guangxi Medical University (Nanning, Guangxi, P.R. China) (approval number 2024-E454-01). The human liver specimens were from patients who underwent liver surgery due to hepatobiliary diseases. Each patient provided written informed consent. Among them, 69 paracancerous liver tissues were from the First Affiliated Hospital of Guangxi Medical University (human cohort 1), The second cohort included 32 human liver specimens diagnosed with MASLD. We excluded patients with viral hepatitis, autoimmune liver disease, and high alcohol intake (210 g per week for men and 140 g per week for women) (Rinella et al, 2024). The paracancerous tissues were obtained at least 2 cm away from the distal end of the para-tumoural area and histologically confirmed as non-tumor liver tissues. The MASLD patients were diagnosed based on histological analysis. Detailed characteristics of the patients are listed in Appendix Tables S1 and S2. Human peripheral blood samples were collected from cohort 1 and 2 patients. Serum insulin levels were analyzed by human insulin ELISA kit (Crystal Chem, #90085). Fasting blood glucose and triglyceride levels were obtained from the hospitalization records. The HOMA-IR and HOMA-β indexes were calculated from the fasting levels of serum glucose (mmol/L) and insulin (μU/ml). HOMA-IR = fasting blood glucose × fasting blood insulin/22.5. HOMA-β = 20 × fasting blood insulin/(fasting blood glucose −3.5). TyG = Ln (fasting serum triglycerides [mg/dL] × fasting serum glucose [mg/dL]/2).

## Generation of hepatocyte-specific *Aspg* knockout mice

The loxP *Aspg* mouse in the C57BL6 background was generated by GemPharmatech that was flanked on exon 2 of *Aspg* with loxP sites. The *Aspg*-floxed mice were crossed with *Alb-Cre* mice to generate liver-specific knockout (LKO) mice. The same sex animals from the same litter were randomly assigned to experimental groups. All animal handling protocols were approved by the Animal Care and Use Committee of the Fudan University Shanghai Medical College (approval number 20250227-048).

## Animal studies

Mice were housed in a 12-h light/dark cycle at a room temperature of 22 °C ± 2 °C, humidity of 50% ± 5%, with free access to food and water. Both male and female WT and LKO mice were fed a HFD for 20 weeks for the functional study. For overexpression and rescue assay, female 16-week-old floxed and LKO mice were intravenously injected with AAV8-TBG-*Aspg* or AAV8-TBG-Vector ($2.5 \times 10^{11}$ viral particles per mouse) followed by 16-week HFD feeding (Research Diet, #D12492). For the therapeutic experiment, *Aspg* was ablated by injecting AAV8-TBG-*Cre* to 8-week-HFD-feeding male *Aspg*<sup>flox/flox</sup> mice ($2.5 \times 10^{11}$ viral particles per mouse), followed by another 12-week HFD feeding. For the *Ptp1b* and *Sepp1* knockdown in vivo, AAV-shRNAs ($2.5 \times 10^{11}$ viral particles per mouse) were infused to 8-week HFD-fed male *Aspg*-overexpressing mice. For the western blot analysis of the insulin signaling pathway, the mice were fasted overnight then refed for 2 h to stimulate insulin secretion, except otherwise indicated. The male leptin-deficient (*ob/ob*) mice were purchased from GemPharmatech (T001461). For the STZ-induced late T2D model, the male C57BL/6 mice were fed with HFD fodder (Research Diet, #D12492) for 10 weeks. After being fasted for 12 h with free access to water, these mice were intraperitoneally injected with STZ (MeilunBio, #MB1227; 35 mg/kg in 0.1 M citrate-buffered saline, pH 4.5) for 7 consecutive days, followed by HFD feeding for another 2 weeks. Then the mice were subjected to the analysis for fasting blood glucose level to make sure they were more than 11.1 mmol/L, which was considered to be T2D mice.

## Detection of metabolic indicators

To perform glucose, insulin or pyruvate tolerance tests (GTT, ITT or PTT), mice were fasted for 16 h before the assay. For GTT, mice were given an intraperitoneal (i.p.) injection of glucose (2 g/kg body weight). For ITT, mice were treated with an i.p. injection of insulin (male 0.75 U/kg body weight, female 0.6 U/kg body weight). For PTT, mice were i.p. injection of pyruvate (0.5 g/kg body weight). Then, the concentration of blood glucose was determined using Roche Accu-Chek blood meters from tail venous bleeds at 0, 15, 30, 60, and 120 min after glucose, insulin or pyruvate injection. For GSIS, serum insulin levels were measured by ELISA (Crystal Chem, #90082) after i.p. injection of a glucose bolus (2 g/kg body weight) into mice deprived of food for 16 h. Hepatic glycogen was extracted and determined using a glycogen assay kit following the instructions (Nanjing Jiancheng Bioengineering Institute, China, #A043-1-1). The levels of serum aspartate aminotransferase (AST) (Roche, #20764949322), alanine aminotransferase (ALT) (Roche, #20764957322), triglycerides (TG) (APPLYGEN, #2024R3KE1025) were detected using commercially-available detection kits. Levels of SELENOP protein in serum or culture medium were measured by ELISA Kit (Sangon BioTech, #D721137-0096).

## Hyperinsulinemic–euglycemic clamp

WT and LKO mice fed HFD for 16 weeks were subjected to a hyperinsulinemic–euglycemic clamp assay (Shanghai Anduo Bio-technology, China). Briefly, mice were allowed to recover for 3 days after surgical catheterization of the right jugular vein. Following a fasting for 5 h, mice were first infused with insulin 300 mU/kg/min

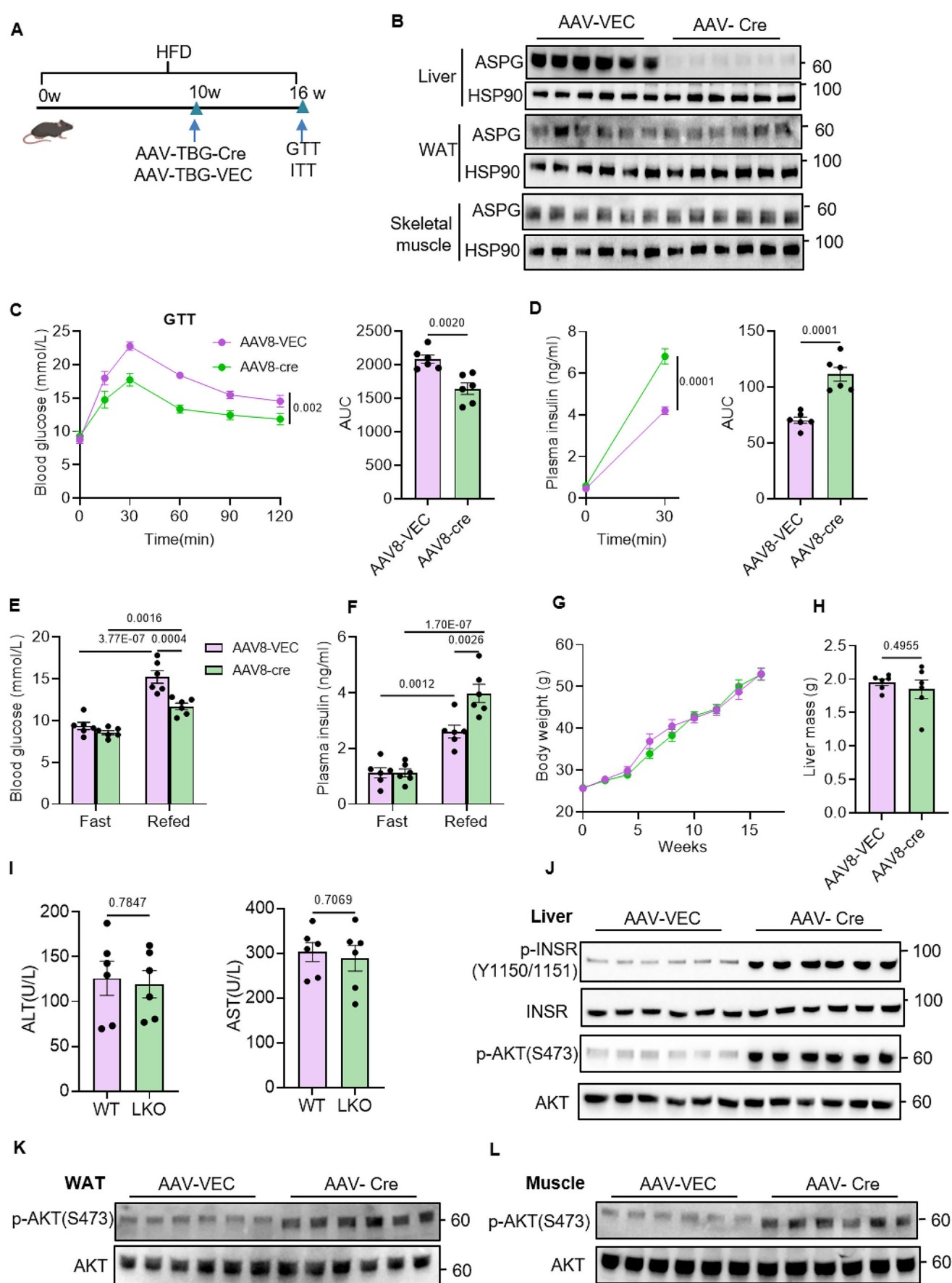

**Figure 8. Targeting hepatic ASPG ameliorates insulin signal transduction defects in obese mice.**

(A) Timeline of mice treatment. Male *Aspg*^flox/flox mice were fed a HFD for 16 weeks. AAV8-TBG-Cre or AAV8-TBG-Vector were injected to mice at the 10th week of HFD feeding. $n = 6$, each group, male. (B) ASPG protein levels in livers, WATs and skeletal muscles in mice of (A). $n = 6$, each group, male. (C, D) GTT (C) and GSIS (D) assay in mice of (A). $n = 6$, each group, male. (E–I) Levels of blood glucose (E), plasma insulin (F), body weight (G), liver mass (H), and serum ALT, AST (I) in mice of (A). Blood glucose and insulin levels were measured after 24 h-fasting or 2 h-refeeding. $n = 6$, each group, male. (J–L) Western blot analysis of the indicated proteins in livers (J), WAT (K), and muscle tissues (L) for mice in (A). $n = 6$, each group, male. Data are represented as mean ± SEM. Statistical analysis was performed by unpaired two-tailed Student's *t* test for (C, D, H, I), by two-way ANOVA followed by Tukey's test for (E, F). Source data are available online for this figure.

within 1 min then with a fixed amount of insulin 2.5 mU/kg/min and a variable amount of 20% dextrose to maintain euglycemia (~7 mM/L) for 120 min. Basal and insulin-inhibited hepatic glucose production were estimated with a continuous infusion of $^{13}$C-glucose for 2 h prior to and throughout the clamp (0.03 mg/kg/min). Blood was sampled from the tip of the tail in heparinized capillary tubes at 0, 90, 100, 110, 120 min for tracer analysis.

## Mouse liver subcellular fractionation

Liver subcellular fractionation was performed as described with minor modifications (Leiro et al, 2023). Briefly, 1.5 g mouse liver was homogenized using a loose-fitting pestle and centrifuged at $600 \times g$ (5 min, 4 °C) twice. The resulting pellet was used to prepare nuclear fraction (Dias et al, 2020). The supernatants were subjected to centrifugation at $9000 \times g$ (10 min, 4 °C) yielding crude mitochondria pellets (reserved for analysis), followed by centrifugation at $20,000 \times g$ (30 min, 4 °C) to pellet plasma membranes. Then after an ultracentrifugation at $100,000 \times g$ (60 min, 4 °C), the endoplasmic reticulum-enriched fraction was pelleted.

## Lipidomics assay

The lipidomics assay was performed by Applied Protein Techonology Inc (Shanghai, China) using UHPLC–MS technology. Briefly, the extracted lipid species from mice livers were separated by ultrahigh performance lipid chromatography (UHPLC Nexera LC-30A, SHIMADZU) followed by mass spectrometry (Q-Exactive Plus Orbitrap system, ThermoFisher) analysis.

## Lysophospholipase activity and PTP1B activity assay

The lysophospholipase activity of mouse ASPG towards 18:0 LPI (Avanti, #850104P), 18:0 LPC (Avanti, #855775P) and 18:0 LPE (Avanti, #856715P) was quantified using HPLC-MS. The lysophospholipid substrate dissolved in DMSO (5 μM in the reaction system) and purified ASPG protein (0.5 μg) were added to the reaction buffer (50 mM Tris–HCl, pH 7.4, containing 0.1% BSA) to give a total volume of 100 μl. The reaction was incubated at 37 °C for the indicated time and was stopped by the addition of chloroform:methanol (2:1 v/v) and 2% orthophosphoric acid. The upper organic phase was separated by centrifugation. The amount of substrate lysophospholipids at each time point was quantified by HPLC-MS (Fudan University, Shanghai, China). The catalytic activity of ASPG was shown as the decrease in LPI amount with time. PTP1B activity was measured using PTP1B Colorimetric Assay Kit (BPS Bioscience, #30019) following the instructions. Briefly, add 10 μl of 1 μM, 5 μM, 10 μM and 50 μM lysophospholipid (including LPI, LPC or LPE) or 10 μl of solvent (DMSO or BSA) to each well. Then, 20 μl PTP1B protein (5 ng/μl) was

loaded to each well. Absorbance at 415 nm of each sample was determined by a multilabel reader after 1 h incubation at room temperature. The PTP1B activity without any lysophospholipids was marked as 1.

## Microscale thermophoresis

MST was performed by Monolith X instrument (NanoTemper, Germany) following the instruction. In brief, purified human His-PTP1B protein was labeled with a red-tris-NTA 2nd generation His-tag labeling kit (NanoTemper, #MO-L018, Germany). The labeled PTP1B protein was then incubated at 50 nM with twofold serial dilutions of ligand LPI (the starting concentration of LPI was 50 μM) in PBS-T buffer (PBS with 0.05% Tween-20). The binding reactions were mixed by pipetting and incubated for 10 min at room temperature, then were loaded into the capillaries (Nano-Temper, #MO-K022, Germany) and measured the samples. The settings are 40% MST power, fluorescence before 5 s, MST on 30 s, fluorescence after 5 s, and delay 25 s. Fnorm = F1/F0 (Fnorm: normalized fluorescence; F1: fluorescence after thermodiffusion; F0: initial fluorescence or fluorescence after T-jump). $K_d$ was determined by the NanoTemper analysis software.

## Histological examination

To perform histologic analysis, the fresh mouse liver or pancreas tissue samples were fixed with 4% paraformaldehyde, embedded in paraffin, and sectioned transversely. The thin sections of liver samples were subjected to Hematoxylin and Eosin staining to observe the cellular morphology or Periodic acid-Schiff staining to visualize glycogen contents in the liver (Shanghai Runnerbio Technology CO., Ltd). The pancreas sections were incubated with anti-glucagon (Cell Signaling Technology #3014 s, 1:10,000) and anti-insulin (Cell Signaling Technology #2760 s, 1:800) antibodies at 4 °C overnight. After washing with PBS, the sections were incubated with fluorescent-conjugated secondary antibodies (Abcam #ab6721, 1:1000). Nuclei were stained with DAPI. All immunofluorescence images were acquired by a laser scanning confocal microscope (Nikon, Eclipse Ci-L).

## Cell culture and treatment

Primary hepatocytes were isolated from 6 to 8-week-old female mice using the collagenase (Sigma-Aldrich Shanghai, #C5138) perfusion method. Isolated hepatocytes, HepG2 and HEK293T were cultured in Dulbecco's modified Eagle's medium (DMEM, MeilunBio) containing 10% fetal bovine serum (FBS, Hyclone) and 1% penicillin–streptomycin. Preadipocytes 3T3L1 were cultured in DMEM containing 10% Newborn Calf Serum (NBCS, Hyclone), 1% Biotin (Sigma-Aldrich, #B4639), pantothenate (Sinopharm

Chemical Reagent Co., Ltd, #67000434) and 1% penicillin–streptomycin. Two days after reaching 100% confluence (designated day 0), preadipocyte 3T3L1 were induced to differentiate into white adipocytes with induction medium (DMEM containing 10% FBS, 0.5 mM 3-isobutyl-1-methylxanthine, 1 μM Dexamethasone and 170 nM insulin) until day 2. Cells were then cultured in DMEM containing 10% FBS and 170 nM Insulin for 2 days. From days 4 to 8, cells were cultured with DMEM containing 10% FBS, which was changed every other day. Usually, the cells were harvested on day 8 for subsequent assays.

All cell lines were confirmed to be mycoplasma-free with a mycoplasma detection kit and treated with Mycoplasma Elimination Reagent for the prevention of mycoplasma contamination. All cells were cultured in a humidified atmosphere containing 5% $CO_2$ at 37 °C. Palmitic acid (PA, 200 μM) (Sigma-Aldrich, #P0500) treatment for 36–48 h was used to induce insulin resistance or lipid accumulation in primary hepatocytes. To stimulate insulin signaling, primary hepatocytes were stimulated with 10 nM insulin for 15 min just before cell harvest. In order to ascertain the impact of LPI on primary hepatocytes, hepatocytes were treated for 12 h with 5 μM of LPI (Avanti Polar Lipids, #850091 P) and/or 10 μM ML-193 (MCE, #HY-110125) for 6 h. To observe the regulation on *Sepp1* expression, primary hepatocytes were treated with 10 nM insulin for 24 h and 5 μM LPI for 12 h with or without 2 μM AKTi (MCE, # HY-10355) for 2 h.

## Immunofluorescence assay

HepG2 cells were plated in 35-mm glass-bottom dishes at 90% confluence. To induce lipid droplet formation, HepG2 cells were treated by 200 μM palmitic acid for 24 h. Then cells were fixed with 4% paraformaldehyde in PBS for 15 min and then washed three times with PBS. Cells were treated with 0.1% Triton X-100 in PBS for an additional 15 min at room temperature to penetrate the cell membrane. After that, cells were blocked with 10% donkey blocking serum before incubation with anti-Flag (Proteintech #20543-1-AP, dilution 1:200) primary antibodies at 4 °C overnight. Then, cells were washed and incubated with anti-rabbit Alexa Fluor 594 secondary antibody (Abcam #ab150080, dilution 1:200) in blocking reagent for 2 h at room temperature. The isotype antibodies were used as the controls. After rinsing with PBS, nuclei were counter-stained by DAPI in PBS for 15 min. Intracellular lipid droplets were stained in a 1 mg/L BODIPY 493/503 solution (ThermoFisher, D3922), or HepG2 cells were incubated for 20 min in 5 μM DiOC6(3) (Shanghai MaoKang Biotechnology Co., Ltd, #MX4009) working solution for ER staining. Images were captured with a Leica DMI6000B microscope.

## RNA extraction and quantitative RT-PCR for gene expression

Total RNA was collected from liver tissues or cultured hepatocytes using Trizol Lysis Reagent (Vazyme, #R401-01). RNA was converted to cDNA using a High-Capacity cDNA Reverse Transcription Kit from Vazyme (#RV101-01). Quantitative PCR was performed with a QuantStudio™ real-time PCR thermal cycler, using SYBR Green Supermix (Vazyme, #Q511-02/03). The $2^{-\Delta\Delta CT}$ method was used to analyze the relative changes in gene expression for qRT-PCR experiments. Relative gene expression to that of the

housekeeping gene *Rplp0* was calculated. Data were normalized and the control group was set at 1.0. Primers are listed in Appendix Table S3.

## Expression and purification of ASPG and PTP1B proteins

Constructed plasmids PET28a-His-*Aspg* and PET28a-His-*Ptp1b* were expressed in E. coli BL21 as an N-terminally positioned His fusion protein. After application of 0.5 mM IPTG for 18 h at 18 °C to induce APSG expression and at 25 °C to induce PTP1B expression, bacterial lysates were incubated with Ni-beads (Smart Lifesciences, #A20091701) for 3 h at 4 °C. Then, Ni-beads were eluted with different concentrations of imidazole solution. The purified ASPG protein was confirmed with the Caucasian blue stain and western blot using an anti-ASPG antibody (Sigma-Aldrich, #SAB1301923). Then it was stored as aliquots at −80 °C until required.

## Adeno-associated virus, adenovirus and lentivirus package

Adeno-associated virus expressing *Aspg*, *Cre*, *Ptp1b shRNA*, *Sepp1 shRNA* were purchased from OBiO Technology (Shanghai) Corp., Ltd. Adenovirus particles were packaged in Ad293T cells and purified by cesium chloride density-gradient ultracentrifugation. Hepatocytes were infected with adenoviruses for 48 h, followed by gene expression analysis. Lentiviral plasmids, psPAX2 and pMD2.G plasmids were co-transfected to HEK293T cells with polyethylenimine (PEI, YEASEN #40815ES03). The culture media were harvested at 72 h post transfection. Centrifuge the viral supernatant at $1000 \times g$ for 5 min to pellet any packaging cells. The supernatant was aliquoted and frozen in −80 °C freezer. For PTP1B knockdown, PTP1B siRNA (20 nM per well) or lentiviral shRNA was transfected or infected into primary hepatocytes in six-well plates. The sequences of shRNA and siRNA are listed in Appendix Table S4.

## Chromatin immunoprecipitation analyses

Chromatin immunoprecipitation (ChIP) was performed as previously described (Zhao et al, 2020). Specifically, $1 \times 10^6$ primary hepatocytes were cross-linked with 1% formaldehyde for 5 min at room temperature. After centrifugation, cell pellets were sonicated on ice to obtain 400–500 bp DNA fragments. The sonicated DNA solution was incubated with 2 μg control IgG (Abcam, #ab199093) or anti-FOXO1 antibody (Cell Signaling Technology, #2880) overnight at 4 °C with rotation. Protein A sepharose slurry was used to collect the antibody–protein–DNA complex by incubation for an hour at 4 °C with rotation. The captured chromatin DNAs were reversely cross-linked by heating overnight at 65 °C. DNA was purified using the Qiagen Qiaquick PCR purification kit, followed by real-time PCR analysis. The primer sequences for the ASPG promoter were listed in the Appendix Table S5.

## Western blotting analysis

Murine liver tissue or cells were homogenized in cell lysis buffer (200 mM NaCl, 50 mM Tris (pH 7.5), 0.5% Triton X-100, 5% Glycerol) with PMSF (YEASEN, #20104ES03) and protease

inhibitor cocktail (YEASEN, #20109ES05) using a bullet blender (Scientific Industries). After centrifuged for 15 min at $12,000 \times g$ at 4 °C, equal amounts of protein were loaded onto a SDS-PAGE gel for separation and transferred to a nitrocellulose membrane (Cytiva, #10600002). Membranes were blocked in a solvent containing PBS/0.1% (v/v) Tween-20 (PBS-T) and probed with primary antibodies in a primary antibody dilution buffer (Beyotime, #P0256). They were then incubated with horseradish peroxidase-conjugated secondary antibody in PBS-T and developed using ECL detection reagent (Sagecretion).

### Dual-luciferase reporter assay

The 1 kb human *Aspg* promoter or 2 kb mouse *Aspg* promoter sequence was cloned into the luciferase reporter vector PGL3 plasmid. Predicted FOXO1-binding sites were mutated (human: GGTTTAC to GGCGCAC around −980 bp, mouse: TGTTTAC to TGCGCAC around −894 bp) to generate mutant pGL3 constructs. The final constructs were validated by DNA sequencing.

Luciferase activity was analyzed using the dual-luciferase reporter assay system (Promega, WI, USA) according to the manufacturer's instructions. Briefly, HK293T cells were seeded in 48-well plates at a number of $1.3 \times 10^5$ cells per well. Luciferase reporter plasmid (100 ng), PCMX-FOXO1 expression plasmid (0, 10, 25, or 50 ng) and pSV40-Renilla plasmid (5 ng) were co-transfected into HK293T cells using the PEI reagent (YEASEN, #40815ES03). The next day, cells were treated with 10 nM insulin for another 24 h, then were harvested. The luciferase activity was measured using a NovoStar Microplate Reader (Ramcon, Denmark). The value was normalized by Renilla luciferase activity.

### Statistical analysis

All values are expressed as the mean ± SEM. We applied a two-tailed, unpaired Student's *t* test for two-group comparison. One- or two-way ANOVA followed by Tukey post hoc tests was used for more than two-group comparisons. $P < 0.05$ was considered statistically significant. All statistical analyses were performed by GraphPad Prism 9.0 software (GraphPad Software, San Diego, CA, USA).

## Data availability

No large-scale data amenable to data repository deposition were generated in this study.

The source data of this paper are collected in the following database record: biostudies:S-SCDT-10_1038-S44318-025-00525-x.

## Peer review information

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

## Acknowledgements

We thank Drs. Xiaoying Li in Fudan University and Yan Lu at Shanghai Sixth People's Hospital for insightful suggestions. This work was supported by the Program of Shanghai Academic Research Leader (22XD1400500 to DP), the National Natural Science Foundation of China (32471356, 32171140 to DP, 82400940 to QZ, 82200981 to YX), and the Special Funds of Taishan Scholars Project of Shandong Province (TSQN202312384 to YX).

## Author contributions

**Feiyan Li**: Data curation; Formal analysis; Investigation; Methodology. **Hua-Sheng Huang**: Resources. **Qingwen Zhao**: Funding acquisition. **Wei Zhang**: Methodology. **Ting Shi**: Investigation. **Wenjing Lv**: Investigation. **Qi Zhu**: Investigation. **Haojie Liu**: Investigation. **Yingjiang Xu**: Funding acquisition. **Haiyan Huang**: Investigation; Methodology. **Qi-Qun Tang**: Supervision. **Yue Gao**: Resources. **Tao Peng**: Resources. **Dongning Pan**: Conceptualization; Supervision; Funding acquisition; Writing—original draft; Writing—review and editing.

Source data underlying figure panels in this paper may have individual authorship assigned. Where available, figure panel/source data authorship is listed in the following database record: biostudies:S-SCDT-10_1038-S44318-025-00525-x.

## Disclosure and competing interests statement

The authors declare no competing interests.

# Expanded View Figures

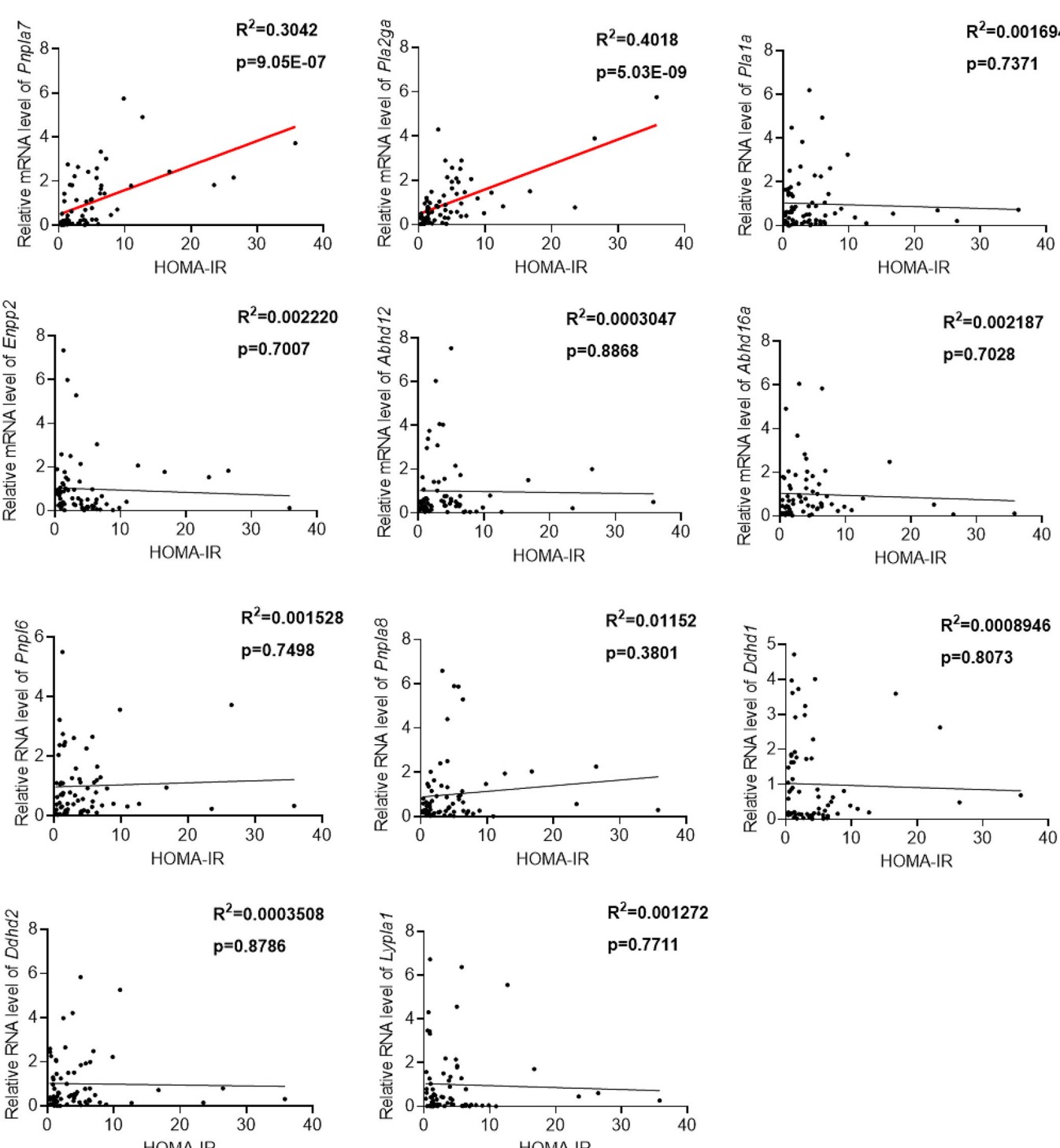

**Figure EV1. The correlation between the mRNA levels of designated (lyso)phospholipases in the human liver samples and HOMA-IR.**

Quantitative RT-PCR detected mRNA levels of designated genes in human liver specimen. $n = 69$, $P$ values examined by the Pearson and Spearman correlation analysis. Source data are available online for this figure.

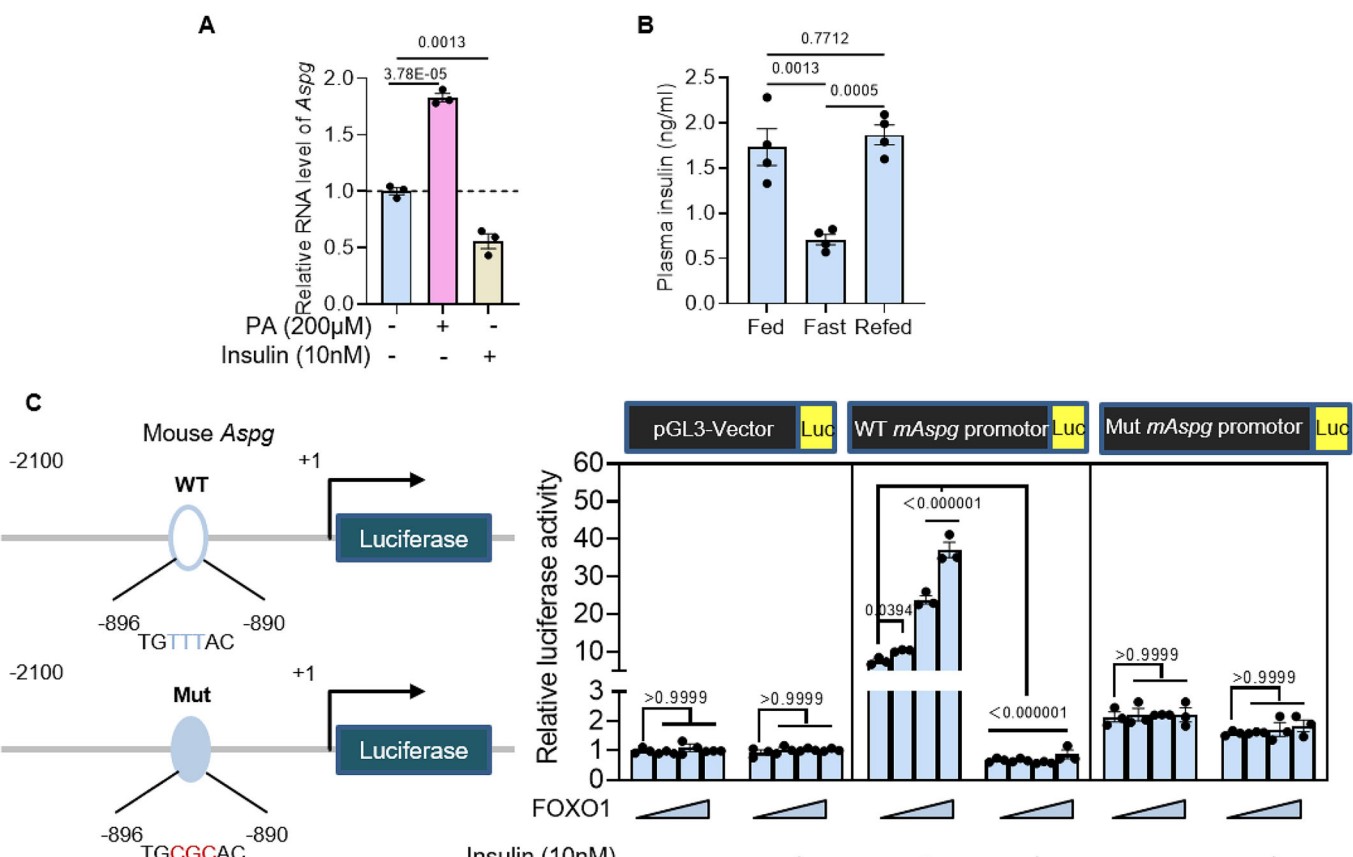

**Figure EV2. Transcriptional factor FOXO1 upregulates hepatic *Aspg* expression.**

(A) Primary hepatocytes isolated from the C57BL/6 mice were treated with 200 μM palmitate (PA) or 10 nM insulin for 24 h. *Aspg* mRNA level was measured by quantitative PCR ($n = 3$). (B) Plasma insulin level in feeding, 24 h-fasting and 2 h-refeeding C57BL/6 mice ($n = 4$). (C) Dual-luciferase reporter assay using mouse *Aspg* WT or mutant promoter to locate the sequence in the *Aspg* promoter responsible for the regulation of FOXO1. HEK293T cells were transfected indicated plasmids and treated with or without 10 nM insulin for 24 h before the assay ($n = 3$). Data are represented as mean ± SEM. Statistical analysis was performed by one-way ANOVA followed by Tukey's test for (A–C). Source data are available online for this figure.

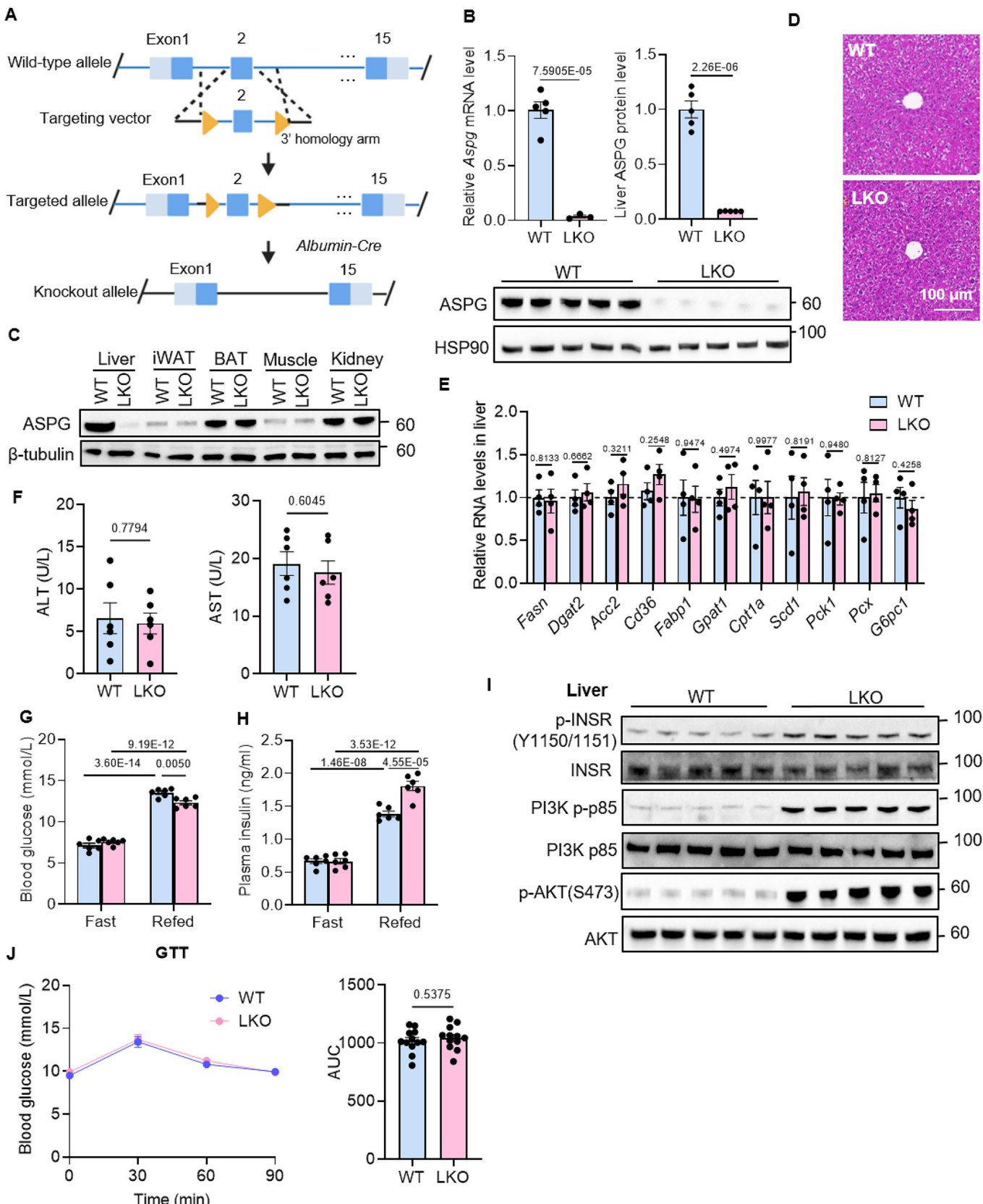

◀ **Figure EV3. The construction of hepatocyte *Aspg* knockout mice.**

(A) Schematic diagram depicting the generation of hepatocyte-deficient *Aspg* mice. (B) Levels of the *Aspg* mRNA and protein in the liver samples from the chow diet feeding WT and *Aspg* LKO mice ($n = 5$ mice per group, male). (C) Representative western blotting showing the expression of ASPG in the liver, inguinal white adipose tissue (iWAT), brown adipose tissue (BAT), skeletal muscle and kidney. (D) The representative images of hematoxylin and eosin staining for liver tissues. Scale bars 100 μm. (E) The mRNA levels of lipogenesis, β-oxidation and gluconeogenesis genes in livers ($n = 4$ each genotype, male). (F) Serum alanine aminotransferase (ALT) (left) and aspartate aminotransferase (AST) (right) levels in chow diet feeding WT and *Aspg* LKO mice ($n = 5$ mice per group, male). (G, H) Levels of blood glucose (G) and plasma insulin (H) were measured in 24 h-fasting and 2 h-refeeding chow diet mice ($n = 6$ each genotype, male). (I) Western blot analysis for the liver tissues from the WT and *Aspg* LKO mice. The mice were fasted for 24 h followed by 2 h-refeeding to stimulate endogenous insulin secretion ($n = 5$ each genotype, male). (J) Glucose tolerance test (GTT) was measured for chow diet feeding mice. ($n = 12$ each genotype, male). Data are represented as mean ± SEM. Statistical analysis was performed by unpaired two-tailed Student's *t* test for (B, E, F, J), by two-way ANOVA followed by Tukey's test (G, H). Source data are available online for this figure.

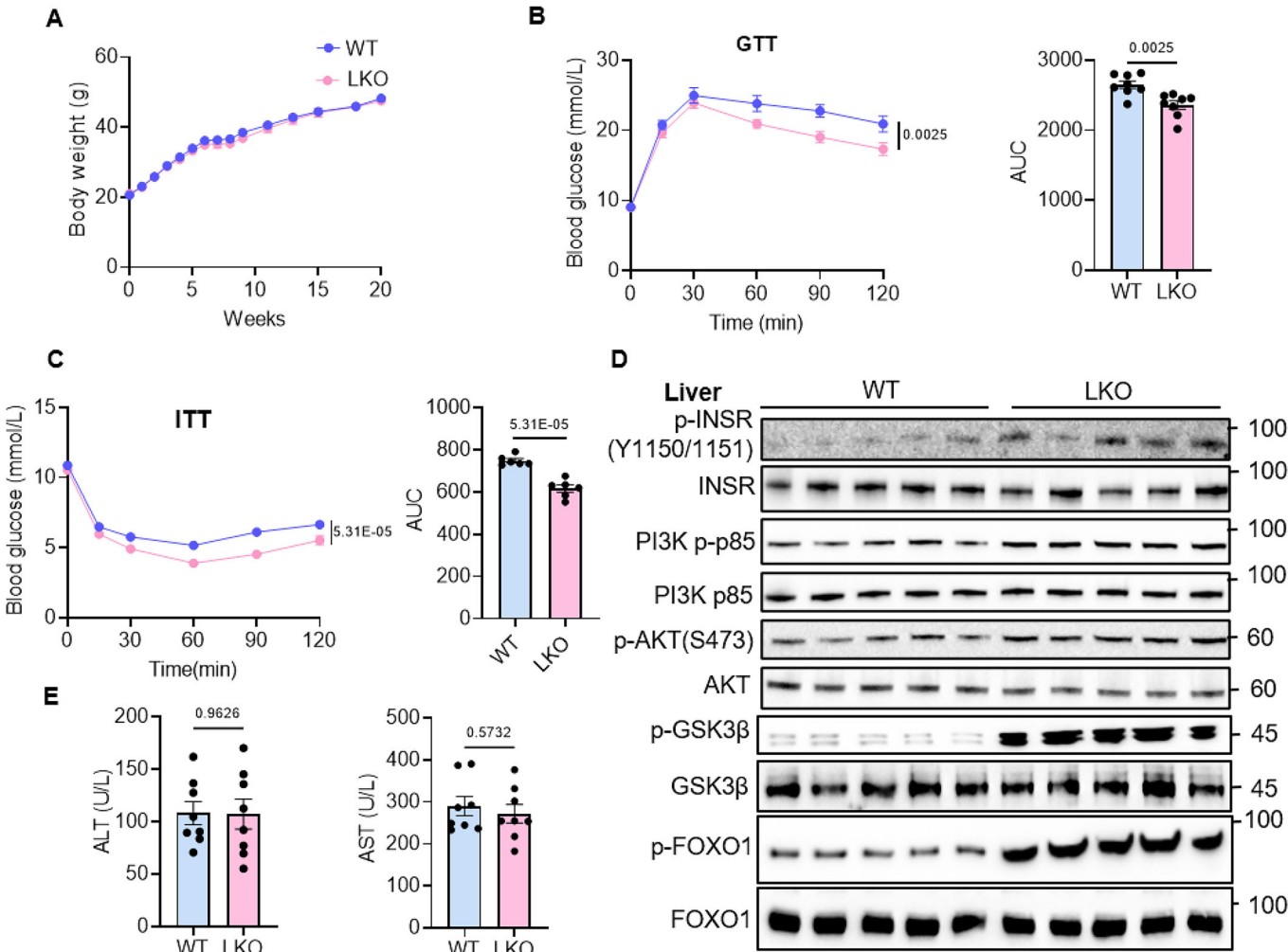

**Figure EV4. Hepatic APSG deficiency improves glucose tolerance in obese female mice.**

(A–E) Comparison of the phenotypic features of the WT and *Aspg* LKO female mice after a HFD for 20 weeks, including (A) body weight, (B) GTT, (C) ITT, (D) western blot analysis of essential markers of insulin signaling in the liver tissues, (E) serum ALT and AST levels ($n = 8$ each group for (A–C) and (E); $n = 5$ each group for (D), female). For all: Data are represented as mean ± SEM. Statistical analysis was performed by unpaired two-tailed Student's *t* test for (B, C, E). Source data are available online for this figure.

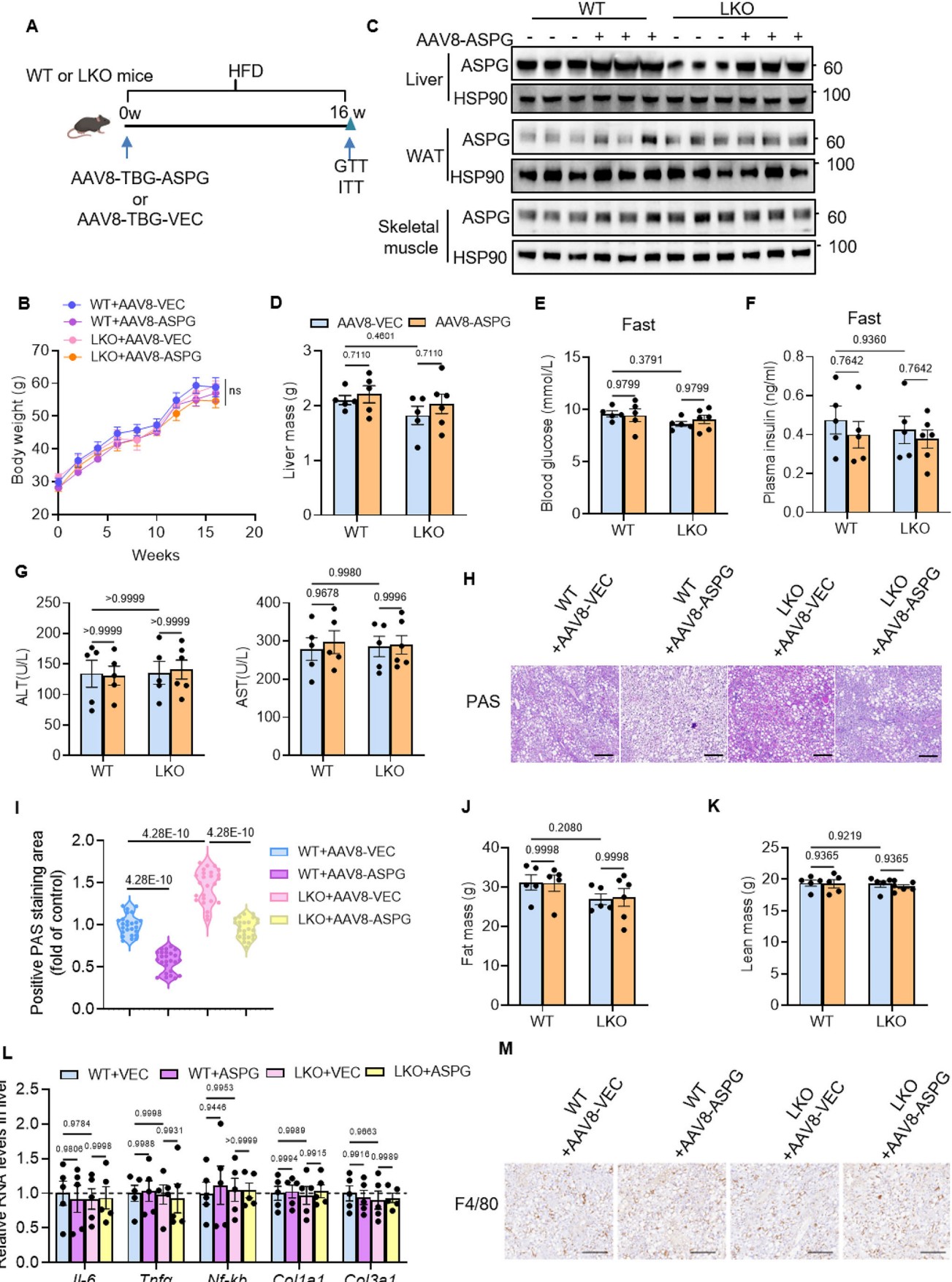

◀ **Figure EV5.  Hepatic overexpression of ASPG does not alter the progression of MASLD.**

(**A**) Timeline of AAV-mediated hepatic *Aspg* expression in female WT and LKO mice. AAV-TBG-ASPG or AAV-TBG-Vector was injected to the WT or LKO mice. Then all the groups of mice were fed a HFD for 16 weeks. (**B**) The body weight of mice in (**A**) during HFD feeding (WT + AAV-VEC $n = 5$, WT + AAV-ASPG $n = 5$, LKO + AAV-VEC $n = 5$, LKO + AAV-ASPG $n = 6$, female). (**C**) Western blot analysis of ASPG expression in the livers, WATs and muscle for mice in (**A**) ($n = 3$). (**D–G**) Liver mass (**D**), fasting blood glucose (**E**) and plasma insulin (**F**) levels, serum ALT and AST levels (**G**) for mince in (**A**) (WT + AAV-VEC $n = 5$, WT + AAV-ASPG $n = 5$, LKO + AAV-VEC $n = 5$, LKO + AAV-ASPG $n = 6$, female). (**H**) The periodic acid-Schiff (PAS) staining showed the glycogen content in livers of mice in (**A**). Scale bar 200 μm. (**I**) Quantification of PAS staining. Total 25 microscope fields of view in each group have been quantified. (**J, K**) Fat mass (**J**) and lean mass (**K**) for mice in (**A**) (WT + AAV-VEC $n = 5$, WT + AAV-ASPG $n = 5$, LKO + AAV-VEC $n = 5$, LKO + AAV-ASPG $n = 6$, female). (**L**) Hepatic mRNA levels of inflammation- and fibrosis-related genes in livers of mice in (**A**) ($n = 5$ each group, female). (**M**) The representative F4/80-immunohistochemistry staining of liver sections for mice in (**A**). Scale bar 200 μm. For all: Data are represented as mean ± SEM. Statistical analysis was performed by two-way ANOVA followed by Tukey's test for (**B, D–G**, and **I–K**), by one-way ANOVA followed by Tukey's test for (**L**). Source data are available online for this figure.

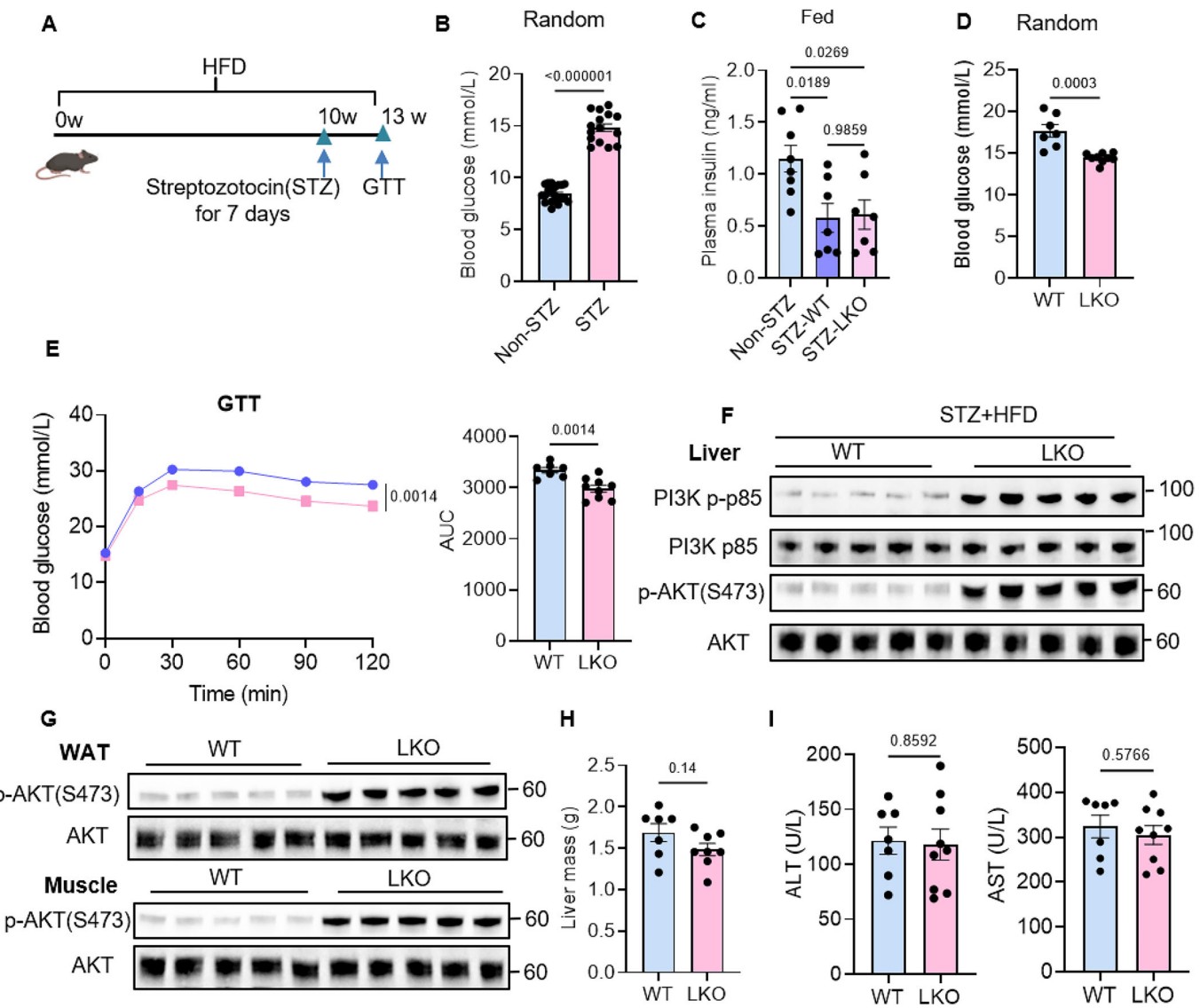

**Figure EV6. Improved glucose tolerance in type 2 diabetic *Aspg* LKO mice.**

(A) Timeline of streptozotocin (STZ)-induced type 2 diabetes mouse modeling. STZ (35 mg/kg body weight) were intraperitoneally injected to the HFD-fed male WT and *Aspg* LKO mice for 7 days. (B, C) Random blood glucose levels (B), Non-STZ $n = 21$, STZ $n = 15$, male) and fed plasma insulin levels (C), Non-STZ $n = 8$, STZ-WT $n = 7$, STZ-LKO $n = 7$, male) were measured in non-STZ, STZ or STZ-WT and STZ-*Aspg* LKO mice. (D) Levels of random blood glucose measured two weeks after STZ injection (STZ-WT $n = 7$, STZ-LKO $n = 9$, male). (E) GTT was performed two weeks after STZ injection (STZ-WT $n = 7$, STZ-LKO $n = 9$, male). The area under the curve (AUC) was used to quantify the GTT results. (F, G) Western blot analysis of essential markers of insulin signaling in the STZ/HFD mice liver, WAT and muscle ($n = 5$ each genotype, male). (H) Liver mass was measured in STZ/HFD mice (STZ-WT $n = 7$, STZ-LKO $n = 8$, male). (I) Serum ALT and AST levels in STZ/HFD mice ((STZ-WT $n = 7$, STZ-LKO $n = 9$, male). For all: Data are represented as mean ± SEM. Statistical analysis was performed by unpaired two-tailed Student's *t* test for (B, D, E, H, I), by one-way ANOVA followed by Tukey's test for (C). Source data are available online for this figure.

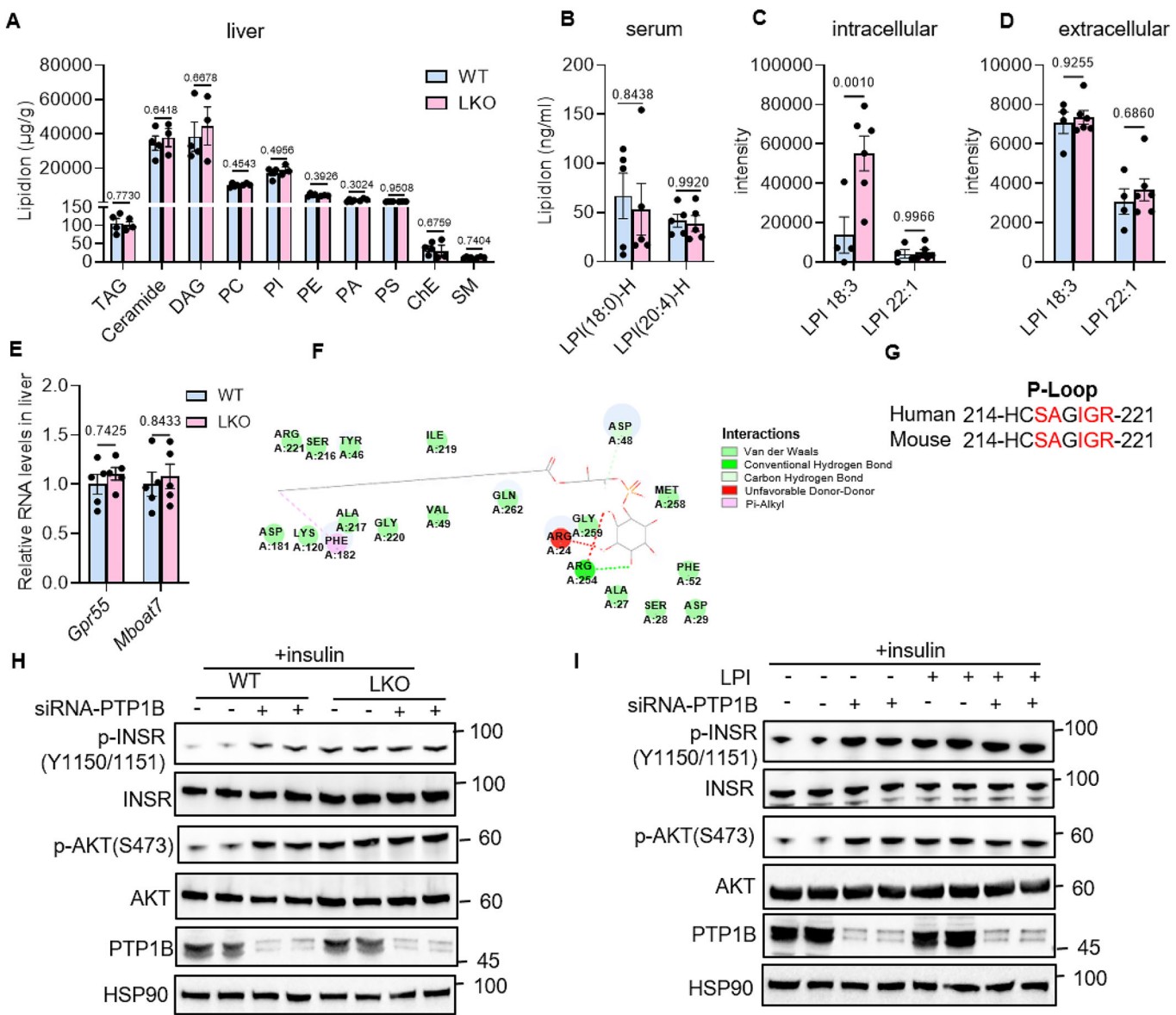

**Figure EV7.  Increased LPIs upon *Aspg* deficiency in hepatocytes inhibit PTP1B activity.**

(A) Quantity of the 10 lipid species in liver tissues from the WT and *Aspg* LKO mice that were fed a HFD for 20 weeks (WT *n* = 4, LKO *n* = 3, female). (B) Quantity of LPIs in serum from the mice treated as in (A) (*n* = 5 each genotype, female). (C, D) The relative quantity of the intracellular (C) and extracellular LPI levels (D) in primary hepatocytes. The hepatocytes were isolated from the chow diet-fed male WT and *Aspg* LKO mice (WT *n* = 4, LKO *n* = 6). (E) Relative mRNA levels of *Gpr55* and *Mboat7* in livers from the WT and LKO mice. The mice were fed a HFD for 16 weeks (*n* = 5 each group, male). (F) Molecular docking revealed interaction of LPI 18:0 with human PTP1B protein. (G) The sequence alignment of human and mouse PTP1B proteins. The amino acid residues interacting with LPI were indicated in red. (H) The siRNA against PTP1B was used to knockdown PTP1B in the primary hepatocytes. Western blot analysis of the indicated proteins was performed after insulin (10 nM) treatment for 15 min. (I) The siRNA against PTP1B was used to knockdown PTP1B in the primary hepatocytes followed by LPI (10 μM) treating the cells for 12 h. Western blot analysis was performed after insulin (10 nM) treatment for 15 min. For all: Data are represented as mean ± SEM. Statistical analysis was performed by unpaired two-tailed Student's *t* test for (A–E). Source data are available online for this figure.

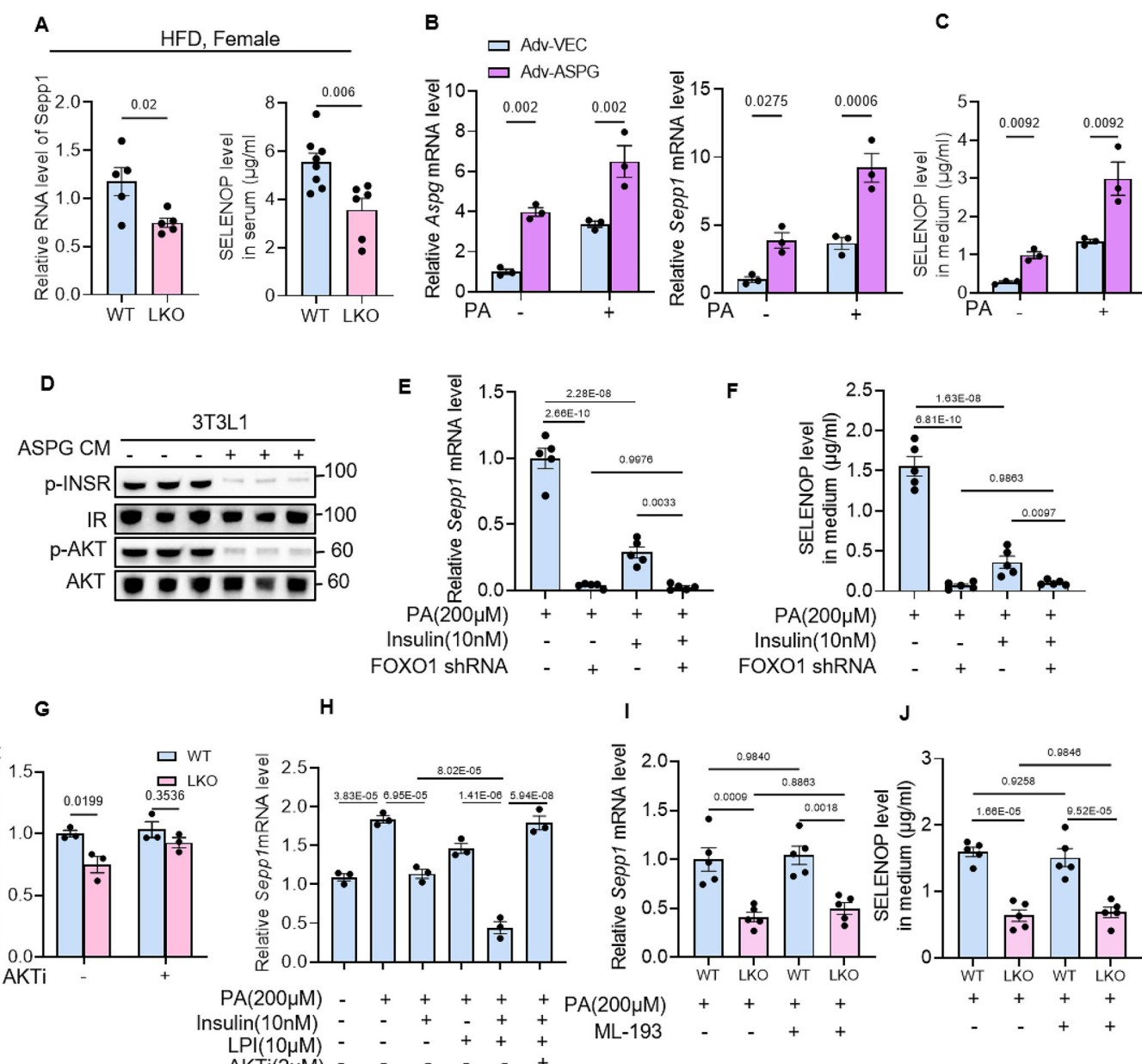

**Figure EV8. ASPG regulates *Sepp1* expression.**

(**A**) Relative mRNA levels of *Sepp1* in livers (left, $n = 5$ female) and SELENOP protein levels in serum (right, WT $n = 8$, LKO $n = 6$, female) from the WT and *Aspg* LKO mice that were fed a HFD for 20 weeks. (**B**) Adenoviruses expressing *Aspg* (Adv-ASPG) or the vector control (Adv-GFP) infected the primary hepatocytes followed by with or without palmitate (200 μM) treating the cells for 24 h. Then the mRNA levels of *Aspg* and *Sepp1* were determined by quantitative real-time PCR ($n = 3$ each group). (**C**) The SELENOP protein levels in the culture medium of cells in (**B**) ($n = 3$ each group). (**D**) Conditional medium collected from primary hepatocytes overexpressing Adv-ASPG after palmitic acid treatment for 24 h was applied to 3T3L1 adipocytes for 48 h. Western blot analysis of the phosphorylation and total protein levels of INSR and AKT in 3T3L1 adipocytes ($n = 3$). (**E, F**) Lentiviral *Foxo1* shRNA infected mouse primary hepatocytes followed by treatment with palmitate (36 h), insulin (24 h). *Sepp1* mRNA (**E**) and SELENOP protein levels (**F**) were measured ($n = 5$ each group). (**G**) Relative mRNA levels of *Sepp1* in the primary hepatocytes isolated from the WT and *Aspg* LKO mice. The cells were treated with or without AKTi (2 μM) for 2 h before the assay ($n = 3$ each group). (**H**) Relative mRNA levels of *Sepp1* in the mouse primary hepatocytes. The cells were treated with palmitate (36 h), insulin (24 h), LPI (12 h) and AKTi (2 h) as indicated following by a quantitative RT-PCR analysis ($n = 3$). (**I, J**) Mouse primary hepatocytes were treated by PA (36 h) and ML-193 (6 h) prior to the *Sepp1* mRNA (**I**) and SELENOP protein assay (**J**). $n = 5$ each group. For all: Data are represented as mean ± SEM. Statistical analysis was performed by unpaired two-tailed Student's *t* test for (**A**), by two-way ANOVA followed by Tukey's test for (**B, C, G**), by one-way ANOVA followed by Tukey's test for (**E, F, H–J**). Source data are available online for this figure.

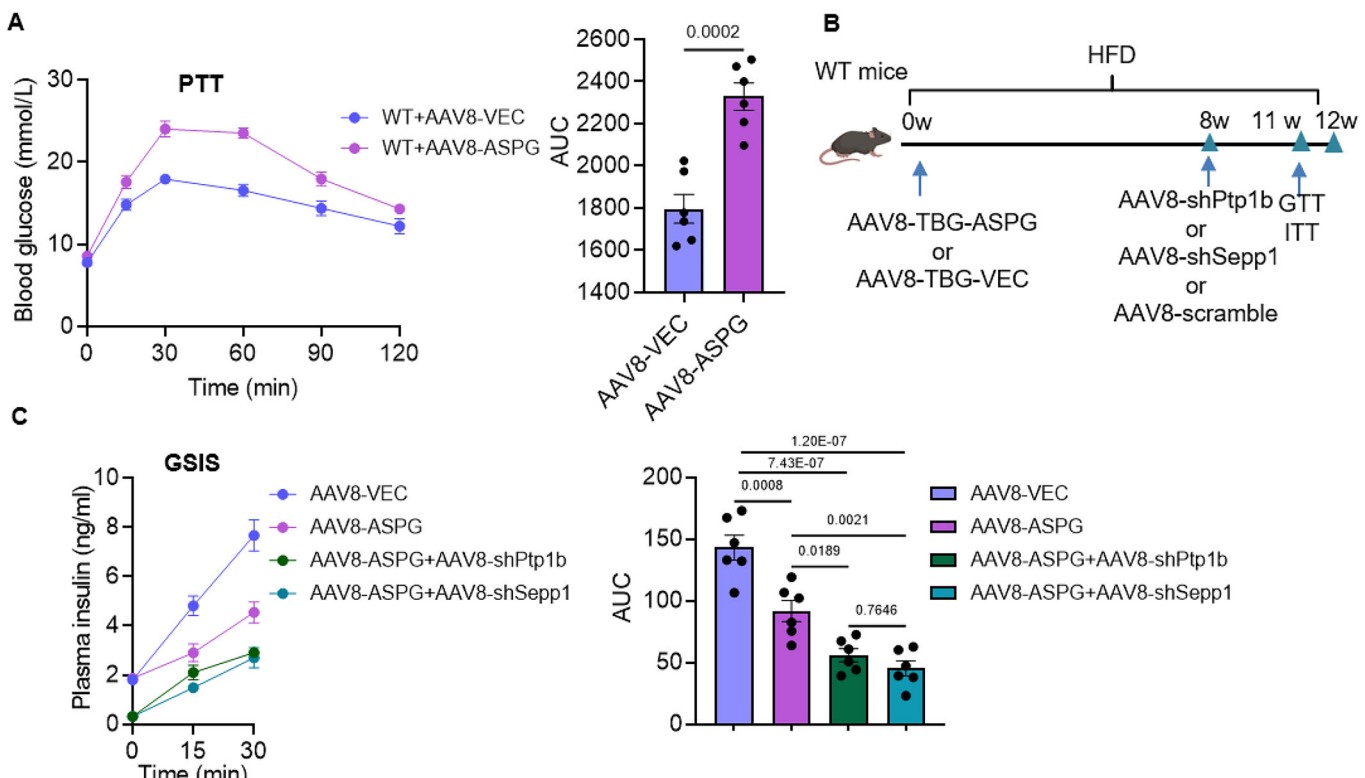

**Figure EV9.  Knocking down *Ptp1b* or *Sepp1* improves glucose metabolic homeostasis in Aspg-overexpression mice.**

(**A**) *Aspg*-overexpressing mice showed increased gluconeogenesis in pyruvate tolerance test (PTT) after 8-week HFD feeding. $n = 6$ each group, male. (**B**) Timeline of AAV-mediated hepatic *Aspg* expression followed by *Ptp1b* or *Sepp1* knockdown. (**C**) The serum insulin levels were measured during the first 30 min of GTT assay. $n = 6$ each group, male. For all: Data are represented as mean ± SEM. Statistical analysis was performed by unpaired two-tailed Student's *t* test for (**A**), by one-way ANOVA followed by Tukey's test for (**C**). Source data are available online for this figure.

