## [Peer Review File · The EMBO Journal]

Hepatic ASPG-mediated lysophosphatidylinositol catabolism impairs insulin signal transduction

Feiyan Li, Hua-Sheng Huang, Qingwen Zhao, Wei Zhang, Ting Shi, Wenjing Lv, Qi Zhu, Haojie Liu, Yingjiang Xu, Haiyan Huang, Qi-Qun Tang, Yue Gao, Tao Peng, and Dongning Pan

Corresponding author: Dongning Pan (dongning.pan@fudan.edu.cn)

Review Timeline:

Submission Date:	19th Dec 24
Editorial Decision:	10th Feb 25
Revision Received:	14th Jun 25
Editorial Decision:	8th Jul 25
Revision Received:	9th Jul 25
Accepted:	17th Jul 25

Editor: Daniel Klimmeck

Transaction Report:

Dear Dr Pan,

Thank you again for the submission of your manuscript (EMBOJ-2024-119990) to The EMBO Journal, as well as providing us with a preliminary point-by-points response to the expert concerns raised. Please accept our apologies for getting back to you with protraction due to delayed referee input, as well as detailed discussion in the editorial team. As mentioned, your study was assessed by three reviewers with expertise in systemic metabolism and insulin signaling, whose comments are enclosed below.

As you will see from their comments, the referees acknowledge the analysis and potential interest and value of your findings. However, they also express major concerns i.p. regarding i) suitability of the assays applied, and conclusive support for core claims made on a role for ASPG in systemic metabolic signaling and insulin production (Ref#2, pts. 3,4; ref#3, pt.1); as well as ii) the degree of mechanistic insights into the connection between ASPG, FOXO1 upstream as well as PTP1B-dependent SELENOP downstream (Ref#1, pt.2,3; ref#2, pt.6; ref#3, pts. 2,3). Referee #1 in addition requests more comprehensive exploration of human conservation of the phenotypes (Ref#1, standfirst). Finally, the reviewers also raise a number of issues related to the data presentation, additional controls and improved methods annotation required, as well as and overall discussion of related literature, that would need to be conclusively addressed to achieve the level of robustness and clarity needed for The EMBO Journal.

Given the overall interest stated and broader angle of your findings, we are able to invite you to revise your manuscript experimentally to address the referees' comments along the lines indicated in the preliminary response. I need to stress though that we do require strong support from the referees on a revised version of the study in order to move on to publication of the work.

Please feel free to contact me if you have any questions or need further input on the referee comments.

When submitting your revised manuscript, please carefully review the instructions below.

Please feel free to approach me any time should you have additional questions related to this.

Thank you for the opportunity to consider your work for publication.

I look forward to your revision.

Kind regards,

Daniel Klimmeck

Daniel Klimmeck, PhD
Senior Editor
The EMBO Journal

Instruction for the preparation of your revised manuscript:

2) individual production quality figure files as .eps, .tif, .jpg (one file per figure).

3) a .docx formatted letter INCLUDING the reviewers' reports and your detailed point-by-point response to their comments. As part of the EMBO Press transparent editorial process, the point-by-point response is part of the Review Process File (RPF), which will be published alongside your paper.

4) a complete author checklist, which you can download from our author guidelines ([https://wol-prod-cdn.literatumonline.com/pb-assets/embo-site/Author Checklist%20-%20EMBO%20J-1561436015657.xlsx](https://wol-prod-cdn.literatumonline.com/pb-assets/embo-site/Author%20Checklist%20-%20EMBO%20J-1561436015657.xlsx)). Please insert information in the checklist that is also reflected in the manuscript. The completed author checklist will also be part of the RPF.

6) It is mandatory to include a 'Data Availability' section after the Materials and Methods. Before submitting your revision, primary datasets produced in this study need to be deposited in an appropriate public database, and the accession numbers and database listed under 'Data Availability'. Please remember to provide a reviewer password if the datasets are not yet public (see <https://www.embopress.org/page/journal/14602075/authorguide#datadeposition>).

7) Our journal encourages inclusion of *data citations in the reference list* to directly cite datasets that were re-used and obtained from public databases. Data citations in the article text are distinct from normal bibliographical citations and should directly link to the database records from which the data can be accessed. In the main text, data citations are formatted as follows: "Data ref: Smith et al, 2001" or "Data ref: NCBI Sequence Read Archive PRJNA342805, 2017". In the Reference list, data citations must be labeled with "[DATASET]". A data reference must provide the database name, accession number/identifiers and a resolvable link to the landing page from which the data can be accessed at the end of the reference. Further instructions are available at .

8) At EMBO Press we ask authors to provide source data for the main and EV figures. Our source data coordinator will contact you to discuss which figure panels we would need source data for and will also provide you with helpful tips on how to upload and organize the files.

Numerical data can be provided as individual .xls or .csv files (including a tab describing the data). For 'blots' or microscopy, uncropped images should be submitted (using a zip archive or a single pdf per main figure if multiple images need to be supplied for one panel). Additional information on source data and instruction on how to label the files are available at .

9) We replaced Supplementary Information with Expanded View (EV) Figures and Tables that are collapsible/expandable online (see examples in <https://www.embopress.org/doi/10.15252/emboj.201695874>). A maximum of 5 EV Figures can be typeset. EV Figures should be cited as 'Figure EV1, Figure EV2' etc. in the text and their respective legends should be included in the main text after the legends of regular figures.

11) For data quantification: please specify the name of the statistical test used to generate error bars and P values, the number (n) of independent experiments (specify technical or biological replicates) underlying each data point and the test used to calculate p-values in each figure legend. The figure legends should contain a basic description of n, P and the test applied. Graphs must include a description of the bars and the error bars (s.d., s.e.m.).

Please remember: Digital image enhancement is acceptable practice, as long as it accurately represents the original data and conforms to community standards. If a figure has been subjected to significant electronic manipulation, this must be noted in the figure legend or in the 'Materials and Methods' section. The editors reserve the right to request original versions of figures and

the original images that were used to assemble the figure.

We realize that it is difficult to revise to a specific deadline. In the interest of protecting the conceptual advance provided by the work, we recommend a revision within 3 months (11th May 2025). Please discuss the revision progress ahead of this time with the editor if you require more time to complete the revisions.

Referee #1:

The manuscript titled "Hepatic ASPG-mediated lysophosphatidylinositol catabolism aggravates systemic insulin resistance" presents a novel finding in the fields of metabolic disease, obesity, and MASLD research. The authors demonstrate that hepatic Asparaginase (ASPG) correlates with human insulin resistance, which is also observed in the MASLD mice model. They provide insightful evidence using an AAV-mediated Aspg ablation in MASLD mice, showing promising beneficial effects on insulin resistance. The primary claim suggests that this study uncovers a novel biological mechanism through which ASPG-LPI modulates MASLD-associated insulin resistance, and ASPG ablation may offer a therapeutic strategy for improving systemic insulin sensitivity. Overall, the text is well-written, and the current study provides important results for the field. Although the data is well-conceived, further experimental evidence, clarification, and human translation would benefit the work. Here are my suggestions for major and minor points that require clarification or revision:

Major points:

1. In the end, if there is hepatic insulin resistance in the KO and that leads to increased islet size remains unclear. The authors refer to (systemic) insulin sensitivity or insulin resistance but ITT remains same. Changes in GTT and altered insulin secretion might indicate a beta cell phenotype. Or perhaps insulin wasn't titrated for the ITT? Hyperinsulinemic-euglycemic clamp studies would be very telling here.
2. Although the authors claim that FOXO1 phosphorylation represses hepatic ASPG expression by insulin treatment, additional data is still needed. As this is one of the major claims of your paper, it would be interesting if the authors could provide a ChIP for checking FOXO1 and ASPG locus interaction in the presence and absence of insulin.
3. The authors decided to profile hepatokine production by mRNA expression. Although it could be related, it is not a direct approach, such as measuring the hepatokine profile by an Elisa assay. Although the possible correlation between gene expression and cytokine production a better experimental approach is necessary to elucidate whether Sepp1 could play a role in the "hepatic ASPG-LPI axis exerted influence on pancreas insulin secretion and insulin sensitivity in adipose tissues and skeletal muscle." Altogether, this is a weaker part of the paper.

Minor Points:

4. How was the mouse number chosen?
5. Please include plasma ALT/AST for all groups.
6. Please provide the original and uncropped WB images.
7. Western blot images (in general): It is well known that ponceau-stained membranes do not provide a fine reference regarding the loading control. HSP90 is also highly questioned due to its variability. Please provide the loading control properly stained for your membranes (i.e., GAPDH, B-tubulin, Vinculin, etc.).
8. Results - Figure 1a - The authors mentioned that they "measured the correlations between homeostasis model assessment of insulin resistance (HOMA-IR) and a panel of (lyso)phospholipases (Figure 1a)". However, Figure 1a is not a correlation analysis, just the panel. As you check for the correlation analysis of HOMA-IR and Aspg mRNA expression, please add this graph to Figure 1a or 1b. In the text, you mentioned that, however, the graph is not presented in Figure 1a, just the panel. I recommend you move it from Figure S1 and add it together with the triglyceride glucose (TyG).
9. Please discriminate the mice's sex. Were the experiments performed in males, females mice, or both?
10. Please be more specific in this sentence: "explored the physiopathological role of ASPG and its substrate lysophospholipids in the control of metabolic homeostasis". What is metabolic homeostasis? Be more specific.
11. Please provide the loading control for Figures 1h and 1i.
12. "The decrease in hepatic glycogen contents as indicated by periodic acid-schiff (PAS) staining further confirmed the aggravation of hepatic insulin resistance upon upregulating ASPG expression in livers (Figure S5h)". Please provide the quantification.
13. The authors claim about the possible interaction between FOXO1 and the hepatokine SELENOP. However, the experiments need further validation and better comprehension to understand their interaction. For example, the authors could perform a FOXO1 knockdown in hepatocytes, and measure Sepp1 gene expression and SELENOP levels in the presence or absence of insulin.
14. "The translational relevance of this fact was indicated by inhibiting ASPG expression in diet-induced steatosis liver to

improve glucose tolerance and insulin secretion. Our work highlights the particular function of intracellular LPI to suppress PTP1B activity, and targeting ASPG-LPI axis is a potential strategy for insulin resistance alleviation in MASLD patients." Please do not translate your current findings to human patients, once it has been passed by previous validation in a translational approach.

Referee #2:

In this manuscript the authors have studied the impact of hepatic ablation of *Aspg*, encoding for Asparaginase, on metabolism in mice. To this end, the authors have studied the phenotype of C57BL6 mice upon ablation of *Aspg* in hepatocytes, and they have measured ASPG expression in human livers. Results indicate that hepatic ASPG levels positively correlates with insulin resistance indexes. Moreover, hepatocyte *Aspg* ablation enhances glucose tolerance and enhanced insulin signaling upon feeding in HFD-treated mice. Moreover, hepatic overexpression of *Aspg* caused glucose intolerance, and reduced plasma insulin levels. A twist in the manuscript is related to the observation that *Aspg* has lysophosphatidase activity that is specific for lysophosphatidylinositol, and the observation that this could be linked to the regulatory function on LPIs as inhibitors of PTP1B, a key element in the insulin signaling pathway. Furthermore, the authors have searched for hepatokines that depend on *Aspg* expression, and they found that hepatic *Aspg* ablation reduced *Sepp1* gene expression that encodes for SELENOP protein. The manuscript is complex, and some of the key aspects of it are not sufficiently documented. Furthermore, the association between *Aspg* ablation and insulin resistance in mouse models is inaccurate.

Major comments.

1. The liver biopsies were obtained from MASLD subjects (32) as well as from subjects bearing liver cancer (69). It is unclear whether the presence of tumors could influence ASPG expression.
2. Figure 1J. Immunofluorescence assays as shown are not robust enough to indicate the precise distribution of ASPG in cells. The authors should perform subcellular fractionation assays.
3. The authors conclude that hepatic ablation of *Aspg* improves insulin sensitivity in mice fed a high fat diet. This conclusion is based on a greater glucose tolerance, normal fasting plasma insulin levels (Figure 3). These data are not sufficient to indicate an enhanced insulin sensitivity, and hyperinsulinemic-euglycemic clamp assays are required.
4. Similarly, the authors conclude that hepatic overexpression of *Aspg* enhanced insulin resistant of HFD-treated mice. However, these mice upon *Aspg* overexpression showed lower insulin levels and higher plasma glucose. This situation can not be explained by an enhanced insulin resistance but to a reduced insulin secretion. Again, this can be appropriately solved by performing hyperinsulinemic-euglycemic clamp studies.
5. The authors show different studies in which they analyzed insulin signaling in livers from control or *Aspg* ablated conditions (phosphorylation of PI3K, AKT, GSK3beta). In these conditions they analyzed mice that were refed for 2 h after an overnight fasting (Figures 3 and 4). This is a very long-time after the onset of insulin action, and it is crucial that the authors analyzed a shorter time points by injection insulin in control and KO mice (5-15 minutes).
6. The concept that the lysophospholipase activity of ASPG regulates insulin signaling through modulation of PTP1B is indeed very interesting. In this connection, there are reports indicating that liver-specific ablation of PTP1B improves glucose homeostasis and lipid profile in mice (Delibegovic et al., 2009). Nevertheless, the authors need to validate this concept by performing rescue studies in *Aspg* knockout mice by overexpressing PTP1B or viceversa by overexpressing ASPG in PTP1B KO mice.

Referee #3:

The present study investigated the role of hepatic asparaginase (ASPG) in the pathogenesis of insulin resistance. The authors found that ASPG expression was strongly correlated with HOMA-IR and MASLD. The authors then showed that ASPG expression is transcriptionally regulated by FOXO1. The liver KO mice showed improved glucose homeostasis and increased insulin signalling in the liver. The authors showed that the mechanisms include 1) increased levels of LPI, which activate GPR35 and also inhibit PTP1B, and 2) decreased expression of the hepatokine SELENOP.

Overall, the present work provides important insights into the role of ASPG in liver function. However, there are several issues that should be carefully addressed by the authors.

1. The authors conclude that loss of ASPG alleviates obesity-associated insulin resistance. However, the data do not support this conclusion as the KO mice showed no change in insulin tolerance. In addition, fasting insulin levels would be expected to decrease if insulin sensitivity were improved; however, the KO mice had higher insulin levels than control mice. These data suggest that the metabolic phenotype of ASPG is more complex than a change in insulin sensitivity. The authors should change the wording to insulin signalling, not insulin sensitivity. In addition, the authors should perform pyruvate tolerance tests and/or hyperinsulinemic-euglycemic clamp. In addition to KO mice, the authors should carefully examine this in ASPG overexpression mice.

2. Mechanistically, the authors concluded that ASPG loss improves insulin signalling by suppressing PTP1B. The authors should demonstrate the functional requirement of PTP1B using inhibitors or genetic knockdown.

3. The authors should discuss or experimentally test the pathways by which loss of ASPG leads to inhibition of SELENOP. For example, does the regulation of SELENOP require GPR55 or PTP1B?

Again, we thank the reviewers for their critical comments and important suggestions that help us to improve the manuscript significantly. All the modifications have been highlighted in red in the manuscript.

Reply to Reviewer #1's comments

1. In the end, if there is hepatic insulin resistance in the KO and that leads to increased islet size remains unclear. The authors refer to (systemic) insulin sensitivity or insulin resistance but ITT remains same. Changes in GTT and altered insulin secretion might indicate a beta cell phenotype. Or perhaps insulin wasn't titrated for the ITT? Hyperinsulinemic-euglycemic clamp studies would be very telling here.

In our manuscript, we describe that the ASPG-LPI axis regulates hepatokine *Sepp1* expression. Then SELENOP alters insulin secretion, and increases insulin resistance in peripheral metabolic tissues (PMID 29162828, 21035759). The metabolic phenotype of hepatic ASPG ablation is complex and entangled alteration in both insulin secretion and insulin sensitivity.

To confirm the *Aspg* ablation enhances the insulin sensitivity, hyperinsulinemic-euglycemic clamp studies were performed in the *Aspg*^{flox/flox} (WT) and *Aspg* knockout mice after a 16-week high-fat diet (HFD) feeding. The glucose infusion rate (GIR) required to maintain euglycemia was considerably higher, and hepatic glucose production (HGP) was more strongly suppressed by exogenous insulin in the *Aspg* LKO mice, indicating an improvement in the insulin resistance (Fig. 3M,N).

Following the suggestion, we also titrated the injected insulin dosage for ITT assay. When insulin was injected at a dose of 0.75U kg⁻¹ body weight for male mice, and 0.6U kg⁻¹ body weight for female mice (note: before we used insulin 1U kg⁻¹ body weight for the ITT assay), the LKO mice demonstrated improved insulin tolerance (Fig. 3K for male mice, Fig. EV4C for female mice), and *Aspg* overexpression aggravated insulin intolerance after HFD feeding (Fig. 7G).

2. Although the authors claim that FOXO1 phosphorylation represses hepatic ASPG expression by insulin treatment, additional data is still needed. As this is one of the major claims of your paper, it would be interesting if the authors could provide a ChIP for checking FOXO1 and ASPG locus interaction in the presence and absence of insulin.

We performed the ChIP assay with an anti-FOXO1 antibody in primary hepatocytes in the presence and absence of insulin (10nM for 2h). As shown in Fig. 2C, FOXO1 binds to the *Aspg* promoter at around -900 region, whereas insulin treatment detached FOXO1 from there.

3. The authors decided to profile hepatokine production by mRNA expression. Although it could be related, it is not a direct approach, such as measuring the hepatokine profile by an Elisa assay. Although the possible correlation between

gene expression and cytokine production, a better experimental approach is necessary to elucidate whether *Sepp1* could play a role in the "hepatic ASPG-LPI axis exerted influence on pancreas insulin secretion and insulin sensitivity in adipose tissues and skeletal muscle." Altogether, this is a weaker part of the paper.

We agree that profiling hepatokines by mRNA expression is not a direct approach. It is possible that deficiency of ASPG does not affect transcription of hepatokine-coding genes. Utilizing quantitative PCR first to screen hepatokines is due to its convenience, economy and efficiency. If we did not get any positive indication, we would like to measure the hepatokine profile by an ELISA assay. Luckily, we found that the mRNA level of hepatokine *Sepp1* was regulated by *Aspg* deficiency or overexpression. Then the SELENOP protein levels in the *Aspg* knockout or overexpression mice serum (Figs. 6A-D, 7A and EV8A) and in cultured medium secreted by primary hepatocytes (Figs. 6G and EV8C,F,J) were verified by ELISA assay.

To verify the functional role of *Sepp1* in the hepatic ASPG-LPI axis, we knocked down *Sepp1* expression in the livers of the *Aspg* overexpression mice. Then GTT, ITT, GSIS and insulin signaling transduction in liver, adipose tissues and skeletal muscle were analyzed (Figs. 7 and EV9). The fasting or postprandial glucose and insulin levels were all dramatically reduced after *Sepp1* knockdown as compared to the *Aspg* overexpression group (Fig. 7B-E). Deficiency of *Sepp1* in hepatocytes also improved glucose and insulin tolerance as indicated by GTT and ITT assay (Fig. 7F,G). Simultaneously, during GTT assay, serum insulin levels were analyzed at 0, 15 and 30 min. The *Aspg*-overexpression mice had declined GSIS capability (Fig. EV9C), possibly owing to ASPG accelerating islet β cell loss as previously shown in Fig. 4E. Intriguingly, mice with *Sepp1* knockdown secreted much less insulin upon glucose stimulus than the *Aspg*-overexpression mice (Fig. EV9C). However, they kept low blood glucose levels during the GTT assay (Fig. 7F), suggesting that a restored systemic insulin sensitivity abrogated the requirement of high insulin secretion. Indeed, liver, adipose tissues and skeletal muscles manifested enhanced insulin signaling transduction upon insulin injection in *Sepp1* knockdown mice (Fig. 7H-J). These *in vivo* experiments further validated that SELENOP played an important role in the ASPG-elicited insulin resistance.

4. How was the mouse number chosen?

The *Aspg*^{flox/flox} mice with or without Albumin-Cre expression were designated as LKO and WT mice in our study. In each experiment, the WT and LKO mice were littermates with the same sex and similar age. For *in vivo* analysis, usually we chose mouse number at least 5 per group and the upper limit depended on the mice availability. For the hyperinsulinemic-euglycemic clamp assay, 5 mice of each genotype were used for the assay. One WT mice died during the assay thus n=4 for the WT mice, n=5 for the LKO mice.

5. Please include plasma ALT/AST for all groups.

We have included plasma ALT/AST levels for all groups. Data were shown in Figs 3F,

3G, 8I, EV3F, EV4E, EV5G, EV6I.

6. Please provide the original and uncropped WB images.

We provided the original and uncropped WB images together with the revised manuscript.

7. Western blot images (in general): It is well known that ponceau-stained membranes do not provide a fine reference regarding the loading control. HSP90 is also highly questioned due to its variability. Please provide the loading control properly stained for your membranes (i.e., GAPDH, B-tubulin, Vinculin, etc.).

Thanks for the suggestion. Now we use β -tubulin as a loading control for Figure 1G to show the ASPG level in diverse mouse tissues, and also for Figs. 1H, 1I and EV3C as requested in point 11.

HSP90 is a molecular chaperone with vital roles in protein folding and maturation. Its conformation, structure, protein interaction profile are strongly influenced by and highly responsive to the cellular environment (PMID 37524848). HSP90 transcription is mainly regulated by temperature. As for other western blot images in our manuscript, we mainly compared the protein profiles between the WT and *Aspg* knockout mouse livers or hepatocytes. In this situation, we would think it is acceptable to use HSP90 as a loading control. We would like to follow the suggestion to carefully choose loading controls for our future study.

8. Results - Figure 1a - The authors mentioned that they "measured the correlations between homeostasis model assessment of insulin resistance (HOMA-IR) and a panel of (lyso)phospholipases (Figure 1a)". However, Figure 1a is not a correlation analysis, just the panel. As you check for the correlation analysis of HOMA-IR and *Aspg* mRNA expression, please add this graph to Figure 1a or 1b. In the text, you mentioned that, however, the graph is not presented in Figure 1a, just the panel. I recommend you move it from Figure S1 and add it together with the triglyceride glucose (TyG).

We are sorry that we did not present the data clearly in Figure 1A. Following the suggestion, we moved the correlation analysis graph between *Aspg* level and HOMA-IR from Figure S1 to Fig. 1A.

9. Please discriminate the mice's sex. Were the experiments performed in males, females mice, or both?

We described the mice's sex in both Methods section and each figure legends. The experiments were performed in both male and female mice (male LKO mice in Fig. 3; female LKO mice in Fig. EV4 and Figure 4; female *Aspg* overexpression mice in Fig. 4, male *Aspg* overexpression mice in Fig. 7).

10. Please be more specific in this sentence: "explored the physiopathological role of ASPG and its substrate lysophospholipids in the control of metabolic homeostasis". What is metabolic homeostasis? Be more specific.

Now we rephased the sentence as “explored the physiopathological role of ASPG and its substrate lysophospholipids in the control of insulin signal transduction”.

11. Please provide the loading control for Figures 1h and 1i.

We used β -tubulin as a loading control for Figure 1H and 1I.

12. "The decrease in hepatic glycogen contents as indicated by periodic acid-schiff (PAS) staining further confirmed the aggravation of hepatic insulin resistance upon upregulating ASPG expression in livers (Figure S5h)". Please provide the quantification.

Figure S5h (Now Fig. EV5H) has been quantified by Image J. Total 25 microscope fields of view in each group have been quantified (Fig. EV5I).

13. The authors claim about the possible interaction between FOXO1 and the hepatokine SELENOP. However, the experiments need further validation and better comprehension to understand their interaction. For example, the authors could perform a FOXO1 knockdown in hepatocytes, and measure *Sepp1* gene expression and SELENOP levels in the presence or absence of insulin.

We knocked down *Foxo1* in primary hepatocytes, and measured *Sepp1* mRNA level and SELENOP levels in the medium by ELISA in the presence or absence of insulin. As shown in Fig. EV8E,F, transfecting *Foxo1* shRNA to hepatocytes diminished *Sepp1* mRNA and protein expression no matter absence or presence of insulin.

14. "The translational relevance of this fact was indicated by inhibiting ASPG expression in diet-induced steatosis liver to improve glucose tolerance and insulin secretion. Our work highlights the particular function of intracellular LPI to suppress PTP1B activity, and targeting ASPG-LPI axis is a potential strategy for insulin resistance alleviation in MASLD patients." Please do not translate your current findings to human patients, once it has been passed by previous validation in a translational approach..

We rephase the wording as: A strong correlation between hepatic ASPG level and insulin resistance index was observed in MASLD patients. In diet-induced steatosis mouse model, inhibiting hepatic ASPG expression improved glucose tolerance and systemic insulin signaling transduction. Our work highlights the particular function of intracellular LPI to suppress PTP1B activity, and targeting ASPG-LPI axis might be a potential strategy for strengthening insulin signal transduction.

Reply to Reviewer #2's comments

1. The liver biopsies were obtained from MASLD subjects (32) as well as from subjects bearing liver cancer (69). It is unclear whether the presence of tumors could influence ASPG expression.

We search the GTEx and TCGA database and compared the *Aspg* level in healthy human liver, hepatocellular carcinoma (LIHC), bile duct cancer (CHOL) and the adjacent non-tumor liver tissues (PMID 32873799, 31730861, 28398314). Generally, the *Aspg* mRNA level is significantly decreased in LIHC and CHOL specimen when compared to their peri-tumor tissues. However, the peri-tumor liver tissues have similar *Aspg* levels as normal livers.

2. Figure 1J. Immunofluorescence assays as shown are not robust enough to indicate the precise distribution of ASPG in cells. The authors should perform subcellular fractionation assays.

We performed subcellular fractionation of mouse liver and detected the ASPG level in endoplasmic reticulum (ER), nucleus, plasma membrane (PM) and mitochondria by Western blot. The results confirmed that ASPG presents in the ER and nucleus fractions (Fig. 1K).

3. The authors conclude that hepatic ablation of *Aspg* improves insulin sensitivity in mice fed a high fat diet. This conclusion is based on a greater glucose tolerance, normal fasting plasma insulin levels (Figure 3). These data are not sufficient to indicate an enhanced insulin sensitivity, and hyperinsulinemic-euglycemic clamp assays are required.

We performed hyperinsulinemic-euglycemic clamp assays in the *Aspg*^{fl^{ox}/fl^{ox}} (WT) and *Aspg* knockout mice after a 16-week HFD feeding. The glucose infusion rate (GIR) required to maintain euglycemia was considerably higher, and hepatic glucose production (HGP) was more strongly suppressed by exogenous insulin in the *Aspg* LKO mice, indicating an improvement in the insulin resistance (Fig. 3M,N).

4. Similarly, the authors conclude that hepatic overexpression of *Aspg* enhanced insulin resistant of HFD-treated mice. However, these mice upon *Aspg* overexpression showed lower insulin levels and higher plasma glucose. This situation can not be explained by an enhanced insulin resistance but to a reduced insulin secretion. Again, this can be appropriately solved by performing hyperinsulinemic-euglycemic clamp studies.

The metabolic phenotypes of hepatic ASPG ablation or overexpression are complex and entangled altered insulin secretion and insulin sensitivity. Hyperinsulinemic-euglycemic clamp is the gold-standard method to assess insulin sensitivity, however, we could not afford to execute the assay in both knockout and overexpression mice due to the high expense.

In our revised manuscript, we presented new data for *Aspg*-overexpressing mice. Mice with hepatic *Aspg* overexpression showed higher postprandial glucose and lower insulin levels after 12-week HFD feeding (Fig. 7C,E). To exclude the potential interference in insulin sensitivity from the different endogenous insulin levels, we performed ITT assay in *Aspg*-overexpressing mice after overnight fasting which would abolish the disparity in serum insulin levels (Fig. 7D). Furthermore, activation of insulin signaling pathway in peripheral metabolic tissues was detected 15 min after insulin administration to overnight-fasted WT and LKO mice. As shown in Fig. 7G-J, *Aspg*-overexpressing mice had higher glucose levels in ITT assay and repressed insulin signal transduction in liver, WATs and muscles. Together with the hyperinsulinemic-euglycemic clamp assay for the *Aspg* LKO mice, we conclude that ASPG could regulate insulin sensitivity.

5. The authors show different studies in which they analyzed insulin signaling in livers from control or *Aspg* ablated conditions (phosphorylation of PI3K, AKT, GSK3beta). In these conditions they analyzed mice that were refed for 2 h after an overnight fasting (Figures 3 and 4). This is a very long-time after the onset of insulin action, and it is crucial that the authors analyzed a shorter time points by injection insulin in control and KO mice (5-15 minutes).

We agree that analyzing insulin signaling in a shorter time points after insulin injection is critical for the insulin sensitivity assay. We analyzed insulin signaling in HFD-fed *Aspg* overexpressing mice 15 min after insulin administration. Liver, adipose tissues and skeletal muscle manifested compromised insulin signaling transduction upon *Aspg* overexpression in the livers (Fig. 7H-J).

For the *Aspg* LKO mice, hyperinsulinemic-euglycemic clamp assays were performed to verify enhanced hepatic insulin sensitivity. Hepatic glucose production (HGP) was more strongly suppressed by exogenous insulin infusion in the *Aspg* LKO mice, indicating an improvement of hepatic insulin sensitivity (Fig. 3N).

6. The concept that the lysophospholipase activity of ASPG regulates insulin signaling through modulation of PTP1B is indeed very interesting. In this

connection, there are reports indicating that liver-specific ablation of PTP1B improves glucose homeostasis and lipid profile in mice (Delibegovic et al., 2009). Nevertheless, the authors need to validate this concept by performing rescue studies in *Aspg* knockout mice by overexpressing PTP1B or vice versa by overexpressing ASPG in PTP1B KO mice.

We utilized AAV-shRNA to knock down *Ptp1b* in HFD-fed *Aspg*-overexpressing mice. Depletion of *Ptp1b* reduced serum SELENOP, glucose and insulin levels, and completely reversed glucose and insulin intolerance observed in the *Aspg*-overexpressing mice (Fig. 7A-G). During GTT assay, we also measured serum insulin levels at 0, 15 and 30 min. The *Aspg*-overexpression mice had declined GSIS capability (Fig. EV9C), possibly owing to ASPG accelerating islet β cell loss as previously shown in Fig. 4E. Intriguingly, mice with *Ptp1b* knockdown secreted much less insulin upon glucose stimulus than the *Aspg*-overexpression mice (Fig. EV9C). Even though, these mice kept low blood glucose levels during the GTT assay (Fig. 7F), suggesting that a restored systemic insulin sensitivity abrogated the requirement of high insulin secretion. Indeed, liver, adipose tissues and skeletal muscles manifested enhanced insulin signaling transduction upon insulin injection in *Ptp1b* knockdown mice (Fig. 7H-J). These *in vivo* data further validated that PTP1B played an important role in the ASPG-elicited insulin transduction defects.

Reply to Reviewer #3's comments

1. The authors conclude that loss of ASPG alleviates obesity-associated insulin resistance. However, the data do not support this conclusion as the KO mice showed no change in insulin tolerance. In addition, fasting insulin levels would be expected to decrease if insulin sensitivity were improved; however, the KO mice had higher insulin levels than control mice. These data suggest that the metabolic phenotype of ASPG is more complex than a change in insulin sensitivity. The authors should change the wording to insulin signaling, not insulin sensitivity. In addition, the authors should perform pyruvate tolerance tests and/or hyperinsulinemic-euglycemic clamp. In addition to KO mice, the authors should carefully examine this in ASPG overexpression mice.

Based on the data we have, we think that ASPG plays a role in regulating both insulin secretion and insulin sensitivity. We fully agree to change the wording insulin sensitivity to insulin signaling. The changes have been highlight in red in the revised manuscript.

We performed hyperinsulinemic-euglycemic clamp assays in the *Aspg*^{flox/flox} (WT) and *Aspg* knockout mice after a 16-week HFD feeding. The glucose infusion rate (GIR) required to maintain euglycemia was considerably higher, and hepatic glucose production (HGP) was more strongly suppressed by exogenous insulin infusion in the *Aspg* LKO mice, indicating an improvement in the insulin resistance (Fig. 3M,N).

Unfortunately, we could not afford to execute the hyperinsulinemic-euglycemic clamp assay in both knockout and overexpression mice due to the high expense. Now, some new data were presented to support the deteriorative insulin sensitivity in the *Aspg*-overexpressing mice (Figs. 7 and EV9). Mice with hepatic *Aspg* overexpression showed higher postprandial glucose and lower insulin levels after 12-week HFD feeding (Fig. 7C,E). To exclude the potential interference in insulin sensitivity from the different endogenous insulin levels, we performed ITT assay in *Aspg*-overexpressing mice after overnight fasting which would abolish the disparity in serum insulin levels (Fig. 7D). Furthermore, activation of insulin signaling pathway in peripheral metabolic tissues was detected 15 min after insulin administration to overnight-fasted WT and LKO mice. As shown in Fig. 7G-J, *Aspg*-overexpressing mice exhibited higher glucose levels in ITT assay and repressed insulin signal transduction in liver, WATs and muscles. All these data support the conclusion that ASPG regulates insulin sensitivity as well as insulin secretion.

2. Mechanistically, the authors concluded that ASPG loss improves insulin signaling by suppressing PTP1B. The authors should demonstrate the functional requirement of PTP1B using inhibitors or genetic knockdown.

We utilized AAV-shRNA to knock down *Ptp1b* in HFD-fed *Aspg*-overexpressing mice. Depletion of *Ptp1b* reduced serum SELENOP, glucose and insulin levels, and completely reversed glucose and insulin intolerance observed in the *Aspg*-overexpressing mice (Fig. 7A-G). During GTT assay, we also measured serum insulin levels at 0, 15 and 30 min. The *Aspg*-overexpression mice had declined GSIS

capability (Fig. EV9C), possibly owing to ASPG accelerating islet β cell loss as previously shown in Fig. 4E. Intriguingly, mice with *Ptp1b* knockdown secreted much less insulin upon glucose stimulus than the *Aspg*-overexpression mice (Fig. EV9C). Even though, these mice kept low blood glucose levels during the GTT assay (Fig. 7F), suggesting that a restored systemic insulin sensitivity abrogated the requirement of high insulin secretion. Indeed, liver, adipose tissues and skeletal muscles manifested enhanced insulin signaling transduction upon insulin injection in *Ptp1b* knockdown mice (Fig. 7H-J). These *in vivo* data further validated that PTP1B played an important role in the ASPG-elicited insulin transduction defects.

3. The authors should discuss or experimentally test the pathways by which loss of ASPG leads to inhibition of SELENOP. For example, does the regulation of SELENOP require GPR55 or PTP1B?

We treated the primary WT and *Aspg* LKO hepatocytes with/without GPR55 antagonist ML193. Loss of *Aspg* led to inhibition of SELENOP expression and secretion. ML193 treatment could not alter SELENOP expression in either the WT or *Aspg*-deficient hepatocytes (Fig. EV8I,J), indicating the independence on GPR55 for ASPG-LPI regulating SELENOP level.

Ptp1b knockdown in both primary WT and *Aspg*-deficient hepatocytes substantially curtailed SELENOP expression and secretion (Fig. 6G). Here, the inhibition effect on *Sepp1* expression by *Ptp1b* knockdown was much more powerful than *Aspg* scarcity. Considering 10 μ M LPI inhibited PTP1B activity by 50% (Fig. 5J), we deduced lack of ASPG only partly suppressed PTP1B activity. So *Ptp1b* siRNA could further decreased *Sepp1* level in *Aspg* knockout hepatocytes. *Ptp1b* knockdown was also accomplished in *Aspg*-overexpressing mice. As expected, depletion of *Ptp1b* reduced serum SELENOP level in the *Aspg*-overexpressing mice (Fig. 7A).

Dear Dr Pan,

Thank you for submitting your revised manuscript (EMBOJ-2024-119990R) to The EMBO Journal, as well for your patience with our feedback. Your amended study was sent back to the referees for their scientific reassessment, and we have received reports from all of them, which I enclose below. As you will see, the other referees state that the work has been substantially enhanced by the revisions and they are now broadly in favour of publication.

Thus, we are pleased to inform you that your manuscript has been accepted in principle for publication in The EMBO Journal.

We now need you to take care of a number of issues related to formatting and data presentation as detailed below, which should be addressed at re-submission.

Please contact me at any time if you have additional questions related to below points.

Thank you for giving us the chance to consider your manuscript for The EMBO Journal. I look forward to your final revision.

Again, please contact me at any time if you need any help or have further questions.

Best regards,

Daniel Klimmeck

>> Author Contributions: Remove the author contributions information from the manuscript text. Note that CRediT has replaced the traditional author contributions section as of now because it offers a systematic machine-readable author contributions format that allows for more effective research assessment. and use the free text boxes beneath each contributing author's name to add specific details on the author's contribution.

More information is available in our guide to authors.
<https://www.embopress.org/page/journal/14602075/authorguide>

>> The manuscript sections should be in the following order: Title page - Abstract & Keywords - Introduction - Results - Discussion - Methods - Data Availability - Acknowledgments - Disclosure Statement & Competing Interests - References - Figure Legends - (Main Tables with legends if applicable) - Expanded View Figure Legends.

>> Please avoid textual redundancy with (Jin et al, 2023; PMID: 38129407) in the Methods section.

>> Figures in separate files: Figures need to be uploaded as individual, high resolution figure files.

>> Please also provide the synopsis figure as a separate high-resolution figure file.

>> Appendix file with ToC: The appendix file needs to be in PDF format; title page should contain "Appendix for + manuscript title"; nomenclature should be Appendix Table Sx throughout manuscript and Appendix PDF.

>> Data availability section: please adjust the title to 'Data availability section' and alter the statement to 'No large-scale data amenable to data repository deposition were generated in this study.'

>> Consider additional changes and comments from our production team as indicated below:

- Data citations: no comments
- Figure Legends (main + EV): Please note that the exact p values are not provided in the legends of figures 2D, EV2 C

Referee #1:

Thank you for considering my points.

You responded that you chose 5 mice per group but I had asked HOW you made that decision. Usually, no one would start with 5 mice per group unless there are good reasons.

Referee #2:

The authors have adequately answered my prior questions and I think the manuscript deserves publication

Referee #3:

The authors provided new data that addressed this reviewer's comments. No further comment from this reviewer.

The authors addressed the remaining editorial issues.

Dear Dr Pan,

Thank you for submitting the revised version of your manuscript. I have now evaluated your amended manuscript and concluded that the remaining minor concerns have been sufficiently addressed.

I am thus pleased to inform you that your manuscript has been accepted for publication in the EMBO Journal.

Related, I would like to hereby ask your consent on keeping the referee response figures included in this file.

On a different note, I would like to alert you that EMBO Press offers a format for a video-synopsis of work published with us, which essentially is a short, author-generated film explaining the core findings in hand drawings, and, as we believe, can be very useful to increase visibility of the work. Please see the following link for representative examples and their integration into the article web page:

<https://www.embopress.org/doi/full/10.15252/embj.2019103932>

Best regards,

Daniel Klimmeck

Daniel Klimmeck, PhD
Senior Editor
The EMBO Journal
EMBO
Postfach 1022-40
Meyerhofstrasse 1
D-69117 Heidelberg
contact@embojournal.org